# On Understanding Attention-Based In-Context Learning for Categorical Data

**Aaron T. Wang** [1] **William Convertino** [1] **Xiang Cheng** [1] **Ricardo Henao** [1] **Lawrence Carin** [1]

## Abstract

In-context learning based on attention models is examined for data with categorical outcomes, with inference in such models viewed from the perspective of functional gradient descent (GD). We develop a network composed of attention blocks, with each block employing a self-attention layer followed by a cross-attention layer, with associated skip connections. This model can exactly perform multi-step functional GD inference for in-context inference with categorical observations. We perform a theoretical analysis of this setup, generalizing many prior assumptions in this line of work, including the class of attention mechanisms for which it is appropriate. We demonstrate the framework empirically on synthetic data, image classification and language generation.

## 1. Introduction

The Transformer architecture (Vaswani et al., 2017) has revolutionized natural language processing, achieving remarkable success in tasks like language generation (Vaswani et al., 2017; Radford et al., 2019; Devlin et al., 2019; Brown et al., 2020; Touvron et al., 2023; DeepSeek, 2025). This success has spurred significant interest in understanding the underlying mechanisms driving Transformer inference, particularly for in-context learning, where the model is presented with a "prompt" of input-output examples and tasked with predicting outputs for new inputs (Liu et al., 2022).

A fruitful line of research has emerged, focusing on a simplified yet insightful paradigm: predicting outputs for a context-dependent *hidden* function. In this setup, the prompt consists of input-output pairs from this function, and the Transformer is queried with a new input. Crucially, the Transformer is *not* explicitly trained to learn this function; instead, it infers the function from the provided context and

predicts the corresponding output. Early work in this direction concentrated on simple function classes, like linear regression (von Oswald et al., 2023; Ahn et al., 2023; Zhang et al., 2024; Schlag et al., 2021), with extensions to nonlinear, kernel-based regression (Cheng et al., 2024) emerging more recently. These studies have successfully linked the inference performed by the Transformer to functional gradient descent (GD) on the latent function.

These analyses have primarily focused on functions with *real-valued* outputs, a significant departure from the *categorical* nature of language data, where outcomes are discrete tokens. This paper bridges this gap, extending the functional GD framework to handle categorical observations. This extension is crucial for bringing the analysis closer to the complexities of language models. Furthermore, while prior work has largely relied on simulated data, we demonstrate the applicability of our framework to real-world problems, including in-context image classification and, importantly, language modeling. For the latter, we leverage recent advances in targeted datasets designed to illuminate Transformer mechanisms (Eldan & Li, 2023), connecting this line of research with the functional data analysis perspective.

### 1.1. Summary of Contributions

1. We analyze in-context learning with categorical observations, offering new insights into language model behavior. Specifically:

   (a) We demonstrate that encoding categorical data (*e.g.*, tokens) as *learned* embedding vectors arises naturally from a GD perspective on inference (Section 2).

   (b) We introduce a new attention-based architecture, employing self-attention and cross-attention *blocks*, that enables *exact* implementation of multi-step GD with categorical data (Section 3).

   (c) We show that next-word-generation can be modelled as a in-context learning problem over categorical observations (see Section 3.6).

   (d) We provide a theoretical analysis showing that, for in-context learning with categorical observations, our GD-based representation is a stationary point of attention-based inference models (Section 5). This theory addresses softmax-based attention, widely used in language models.

2. We empirically validate our framework through experi-

[1]Electrical & Computer Engineering Dept., Duke University, Durham, NC, USA. Correspondence to: Lawrence Carin <lcarin@duke.edu>.

ments on diverse datasets:

(a) We tackle in-context image classification on ImageNet (Russakovsky et al., 2014), demonstrating the model's ability to handle high-dimensional data and adapt to unseen image categories (Section 6.2).

(b) We apply our GD-based model to language generation, training on a combined corpus of *Tiny Stories* and *Children Stories* (Eldan & Li, 2023). We perform a quantitative analysis using GPT-4o scoring (Section 6.3), and demonstrate that the GD-based model yields similar performance as a comparable Transformer, with far fewer parameters.

Our principal motivation is to gain insight into how the Transformer performs in-context learning when the observations are categorical. It is hoped that such insights are relevant to mechanisms involved in the success of Transformers for language modeling. Additionally, the model we have developed may be of interest in its own right, for applications in few-shot learning with categorical outcomes.

### 1.2. Related Work

The broader context of our research is few-shot learning, where models are trained to generalize from limited examples (Schmidhuber, 1987; Bengio et al., 1992). In this context, meta-learning aims to learn a "learning algorithm" that can quickly adapt to new tasks (Andrychowicz et al., 2016; Ravi & Larochelle, 2017; Finn et al., 2017). The Transformer, with its ability to perform in-context learning without parameter updates, can be viewed as a powerful meta-learner (Brown et al., 2020).

A growing body of work specifically investigates the in-context learning capabilities of Transformers. One line of inquiry focuses on characterizing the types of functions that Transformers can effectively learn from context, ranging from simple linear functions (von Oswald et al., 2023; Ahn et al., 2023; Zhang et al., 2024; Schlag et al., 2021) to more complex nonlinear functions (Cheng et al., 2024) and even Bayesian models (Muller et al., 2022; Garg et al., 2022).

Our work is closely aligned with research that interprets Transformer inference as a form of functional gradient descent. This perspective has been successfully applied to analyze in-context learning in settings where the target function is linear (von Oswald et al., 2023; Ahn et al., 2023) or resides in a reproducing kernel Hilbert space (RKHS) (Cheng et al., 2024).

The connection between attention mechanisms and kernels has been explored in various contexts. Linear attention can be seen as a special case of kernel-based attention, where the kernel is simply the inner product (Schlag et al., 2021; von Oswald et al., 2023; Ahn et al., 2023; 2024). More general kernel-based attention allows for richer representations and has been investigated for real-valued data (Cheng et al.,

2024). These studies primarily consider real-valued outputs, while our work extends this framework to the more challenging case of categorical outputs, while also generalizing the form of attention that may be considered.

## 2. Model Setup

### 2.1. Softmax model for token probabilities

Transformer-based in-context learning with categorical data may be viewed from the perspective of two models: ($i$) the Transformer $T_\theta(z)$ with parameters $\theta$ and input $z$, and ($ii$) the model:

$$p_\phi(Y = y | X = x) = \frac{\exp\left(w_{e,y}^T f_\phi(x)\right)}{\sum_{c=1}^C \exp\left(w_{e,c}^T f_\phi(x)\right)}, \quad (1)$$

a softmax representation for the probability of category $y \in \{1, \ldots, C\}$ for covariates $x \in \mathbb{R}^d$, where the context-dependent function $f_\phi(x)$ characterizes covariate dependence. This paper seeks to examine how a gradient-descent (GD) perspective on in-context inference of $f_\phi(x)$ may be implemented in the forward pass of a Transformer $T_\theta(z)$. To be explicit, we will primarily represent $T_\theta(z)$ in terms of attention-based layers with skip connections.

The models $T_\theta(z)$ and $p_\phi(Y|X)$ are distinct, but they have close interrelationships. Specifically, they *share* parameters $\{w_{e,c}\}_{c=1,C}$, where each $w_{e,c} \in \mathbb{R}^{d'}$. As developed below, the input encoding to $T_\theta(z)$ employs $\{w_{e,c}\}_{c=1,C}$. Further, in its forward pass $T_\theta(z)$ computes $f_\phi(x)$ based on contextual data encoded in $z$, and it then feeds the inferred $f_\phi(x)$ into $p_\phi(Y = y | X = x)$ to predict the token/category of a query. Therefore, the softmax representation in (1) composes the output element of $T_\theta(z)$.

### 2.2. Functional GD for *latent* function input to softmax

The input $z$ to $T_\theta(z)$ is an encoded representation of the context $\mathcal{C} = \{(x_i, y_i)\}_{i=1,N}$ and the query $x_{N+1}$. We seek to understand $T_\theta(z)$ from the perspective of functional GD. In that context, function $f_\phi(x)$ is assumed to reside in functional class $\mathcal{F}$, and following (Cheng et al., 2024) we assume $\mathcal{F}$ is a reproducing kernel Hilbert space (RKHS) (Schölkopf & Smola, 2002). Specifically, consider $f_\phi(x) = A\psi(x) + b$, with $\psi(x)$ a *fixed* mapping of covariates $x$ to a Hilbert space, and the parameters acting in that space are $\phi = (A, b)$. If the dimension of the vector output from $\psi(x)$ is $m$ (which could be infinite, in principle), then $A \in \mathbb{R}^{d' \times m}$, while $b \in \mathbb{R}^{d'}$. As discussed below, while GD is performed on $(A, b)$, the update of these parameters is not performed explicitly, rather the function $f_{\phi_k}(x)$ is estimated directly, where $\phi_k$ represent the parameters at GD step $k$.

The matrix of embedding vectors $W_e \in \mathbb{R}^{d' \times C}$ is *learned*, where column $c$, $w_{e,c} \in \mathbb{R}^{d'}$, represents the embedding vector for category $c \in \{1, \ldots, C\}$ (details in Section 4). When performing inference of $f_\phi(x)$ for new contextual data $\mathcal{C}$ (after the attention model is trained), $W_e$ is fixed.

The cross-entropy loss we wish to minimize at inference, to infer $\phi = (A, b)$ based on context $\mathcal{C} = \{(x_i, y_i)\}_{i=1,N}$, is

$$\mathcal{L}(\phi) = -\frac{1}{N} \sum_{i=1}^{N} \log \left[ \frac{\exp\left[w_{e,y_i}^T(A\psi(x_i) + b)\right]}{\sum_{c=1}^{C} \exp\left[w_{e,c}^T(A\psi(x_i) + b)\right]} \right],$$

As derived in Appendix B, for GD step $k+1$ the following functional update rule is realized for $f_{\phi_{k+1}}(x_j)$:

$$f_{j,k+1} = f_{j,k} + \Delta f_{j,k} \tag{2}$$

$$\Delta f_{j,k} = \frac{\alpha}{N} \sum_{i=1}^{N} \left[ w_{e,y_i} - \mathbb{E}(w_e)_{|f_{i,k}} \right] \kappa(x_i, x_j) \tag{3}$$

where $\mathbb{E}(w_e)_{|f_{i,k}} = \sum_{c=1}^{C} w_{e,c} p_{\phi_k}(Y = c|X = x_i)$. The kernel is represented as $\kappa(x_i, x_j) = \psi(x_i)^T \psi(x_j)$, which is endowed with a reproducing property (Wainwright, 2019). For simplicity, in (3) and in subsequent discussion we ignore the impact of the bias $b$ (the details of its inclusion are provided in Appendix B). The parameters are initialized at step $k = 0$.

As developed further below, the kernel $\kappa(x_i, x_j)$ in (3) manifests an attention weight between positions $i$ and $j$. The requirement that $\kappa(x_i, x_j) = \psi(x_i)^T \psi(x_j)$ for some $\psi(x)$ places restrictions on the form of $\kappa(x_i, x_j)$ (e.g., symmetry), which are inconsistent with softmax-based attention in most Transformer implementations. In Section 5 we extend the above formalism, to also allow softmax-based attention. We also show there that the GD-based attention model (developed in Section 3) is a stationary point of a more general construction.

### 2.3. Connections to prior GD Transformer research
In most prior research motivated on understanding the fundamentals of the Transformer based on a functional GD perspective (von Oswald et al., 2023; Ahn et al., 2023; Mahankali et al., 2024; Cheng et al., 2024), the observations $y_i$ were real-valued vectors. In such a setup $p(Y = y_i|X = x_i) = \mathcal{N}(f(x_i), \sigma^2 I_{d'})$, where the variance $\sigma^2$ is assumed constant (and not estimated) and $I_{d'}$ is the $d' \times d'$ identity matrix (here the observations $y_i \in \mathbb{R}^{d'}$). As detailed in Appendix G, by assuming a model $f_\phi(x) = A\psi(x) + b$ within $\mathcal{N}(f_\phi(x_i), \sigma^2 I_{d'})$, GD yields an update equation identical to (3), with the change $w_{e,y_i} - \mathbb{E}(w_e)_{|f_{i,k}} \to y_i - \mathbb{E}(Y)_{|f_{i,k}}$. Because of the underlying Gaussian model for $Y$, with mean $f_\phi(x)$, i.e., $\mathbb{E}(Y)_{|f_{i,k}} = f_{i,k}$. This property allows multiple steps of GD with real vector observations to be performed exactly with multiple steps of kernel-based self attention (plus skip connections) (Cheng et al., 2024), as the underlying expectation $\mathbb{E}(Y)_{|f_{i,k}}$ is just the function itself.

For categorical observations of interest here, $\mathbb{E}(w_e)_{|f_{i,k}}$ is a *nonlinear* function of $f_{i,k}$, and the embedding vector $w_{e,y_i}$ encodes observed $y_i$. Consequently, multiple steps of GD in this setting *cannot* be performed in terms of multiple self-attention layers alone; a mechanism is needed to address the

nonlinear $\mathbb{E}(w_e)_{|f_{i,k}}$. This motivates the new architecture developed in Section 3, in which self attention is complemented by a form of cross attention.

### 2.4. Natural role of learned token embedding vectors
If the matrix of parameters $A$ is initialized with all-zeros (and the bias $b$ is all-zeros), then the function at each position is initialized as $f_{i,0} = 0_{d'}$, where $0_{d'}$ is an all-zeros $d'$-dimensional vector. With this initiation, $\mathbb{E}(w_e)_{|f_{i,0}} = \frac{1}{C} \sum_{c=1}^{C} w_{e,c}$, i.e., the average of the $C$ learned embedding vectors (subsequently denoted $\bar{w}_e$).

Hence, from (3), the first functional update $\Delta f_{j,0}$ is a kernel-weighted average of inputs $\{(w_{e,y_i} - \bar{w}_e)\}_{i=1,N}$. If $\bar{w}_e = 0_{d'}$, this corresponds exactly to the widely used input encoding of tokens/categories in terms of the associated embedding vectors (Vaswani et al., 2017). This demonstrates that the ubiquitous means of encoding language tokens, in terms of learned embedding vectors, is consistent with a GD perspective on the attention-based inference of $f_\phi(x)$.

## 3. Attention-Based Exact Multi-Step GD
### 3.1. Data encoding
The proposed architecture employs attention *blocks*, with each block composed of a self-attention layer followed by a cross-attention layer. Let $e_{i,k}$ represent the vector input to attention block $k+1$. As depicted in Figure 1, for (3) we consider

$$e_{i,k} = (x_i, w_{e,y_i}, \mathbb{E}(w_e)_{|f_{i,k}}, f_{i,k})^T. \tag{4}$$

Within $e_{i,k}$, the positions of $\mathbb{E}(w_e)_{|f_{i,k}}$ and $f_{i,k}$ may be viewed as "scratch space" where these functions are sequentially updated, to implement GD. The importance of scratch space for Transformer implementation of GD was first noted in Akyurek et al. (2022) (see Sec. C.4.1 in the Appendix of that paper). Like in prior attention-based implementations of GD (von Oswald et al., 2023; Ahn et al., 2023; Mahankali et al., 2024; Cheng et al., 2024), the latent function is updated at each self-attention layer, as $f_{i,k} = \sum_{k'=0}^{k-1} \Delta f_{i,k'}$ for $k \geq 1$, with $\Delta f_{i,k'}$ defined in (3), and $f_{i,0} = 0_{d'}$. Our novelty concerns development of a cross-attention mechanism for updating the nonlinear expectation $\mathbb{E}(w_e)_{|f_{i,k}}$ within each attention block.

Within each attention block, the self-attention layer is composed of *two* attention heads (left and center in Figure 1), and the cross-attention layer by a single attention head (right in Figure 1). The next two subsections summarize each of these layers, and Appendix B provides details of all Transformer parameters.

### 3.2. Self-attention layer
#### 3.2.1. SELF ATTENTION FOR $f_{i,k} \to f_{i,k} + \Delta f_{i,k}$

For this self-attention head, consider key and query matrices $W_k = W_Q$ for which $W_K e_{i,0} = (x_i, 0_{3d'})^T$ and

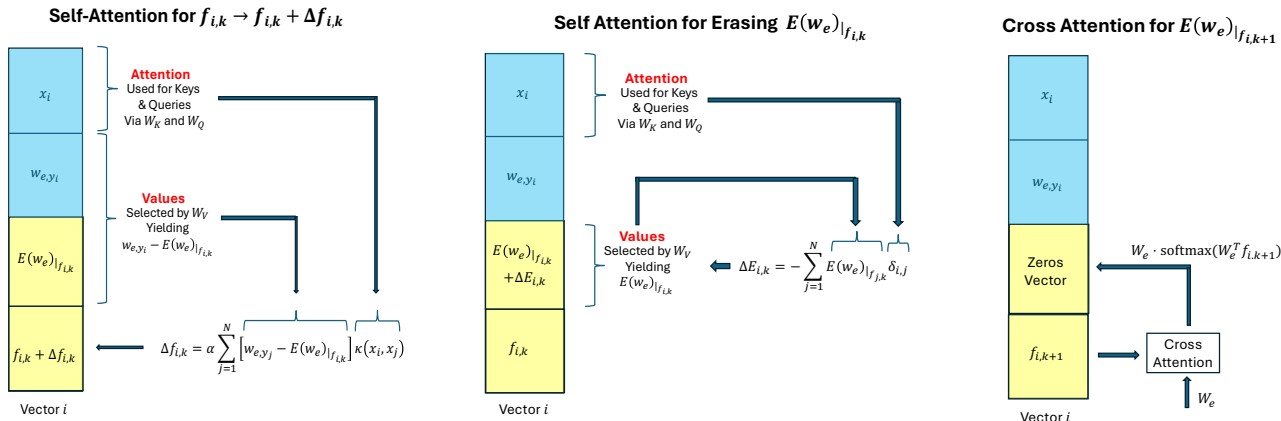

*Figure 1.* The left and middle figures depict the two self-attention heads, while the right-most figure reflects cross attention. A representative vector is shown at position $i$. The portion of the vector in blue remains the same at each layer $k$, and the portion in yellow is updated at each attention block. The sum of $\Delta f_{i,k}$ and $\Delta E_{i,k}$ at left and center is manifested via the skip connection. The form of the vector shown at right, for input to cross attention, depicts the updated $f_{i,k+1}$, and the erased expectation, these the result of the preceding self attention. Cross attention (right) computes the expectation $\mathbb{E}(w_e)_{|_{f_{i,k+1}}}$ and places it at its location within the vector (addition via skip connection). One complete attention block, with skip connections, is depicted in Figure 2.

$W_Q e_{j,0} = (x_j, 0_{3d'})^T$, respectively, and from which kernel-based attention yields $\kappa(W_K e_{i,0}, W_Q e_{j,0}) = \kappa(x_i, x_j)$. These are the same types of key and query matrices as considered previously for this line of research for *real* observations (von Oswald et al., 2023; Ahn et al., 2023; Mahankali et al., 2024; Cheng et al., 2024). The keys are applied to inputs $i = 1, \ldots, N$ and the queries to $j = 1, \ldots, N + 1$. The value matrix $W_V$ is designed to achieve $W_V e_{i,0} = (0_{2+2d'}, w_{e,y_i} - \mathbb{E}(w_e)_{|_{f_{i,k}}})^T$.

Using $W_K$, $W_Q$ and $W_V$ on inputs $\{e_{i,0}\}_{i=1,N}$, for queries $j = 1, \ldots, N + 1$, attention yields $W_V \sum_{i=1}^{N} e_{i,0} \kappa(x_i, x_j)$. The $N + 1$ output vectors tied to the queries are multiplied by a projection matrix $P \in \mathbb{R}^{(d+2d') \times (d+2d')}$, $PW_V \sum_{i=1}^{N} e_{i,0} \kappa(x_i, x_j)$, where here $P$ is just the identity matrix. The resulting vector from this head is

$$\tilde{e}_{i,k}^{H1} = (0_{d+2d'}, \frac{\alpha}{N} \sum_{j=1}^{N} [w_{e,y_j} - \mathbb{E}(w_e)_{|_{f_{i,k}}}] \kappa(x_i, x_j))^T,$$

where $H1$ denotes self-attention-head one (left in Figure 1). Note that this computes $\Delta f_{i,k}$, as defined in (3).

### 3.2.2. SELF ATTENTION FOR ERASING $\mathbb{E}(w_e)_{|_{f_{i,k}}}$

The second self-attention head is used to erase $\mathbb{E}(w_e)_{|_{f_{i,k}}}$. This erasure is necessary in preparation for the subsequent cross-attention layer, where the expectation $\mathbb{E}(w_e)_{f_{i,k+1}}$ is computed, and placed in the location of the erasure.

For this self-attention head, $W_Q$ and $W_K$ are designed such that $W_Q = W_K$ and $W_K e_{i,k} = \lambda(x_i, 0_{3d'})^T$ with $\lambda \gg 0$ a large real number. With large $\lambda$ and using a softmax or radial basis function attention kernel, $\kappa(W_Q e_{j,k}, W_K e_{i,k}) = \kappa(\lambda x_j, \lambda x_i) \approx \delta_{i,j}$, where $\delta_{i,j}$ is

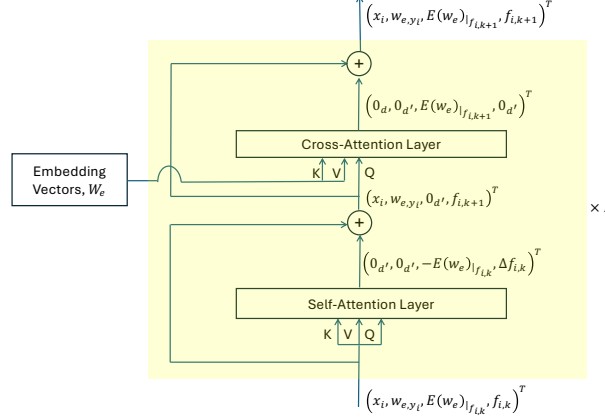

*Figure 2.* Summary of interleaved attention for multi-step GD with categorical observations.

the Kronecker delta function. Further, $W_V$ is set such that $W_V e_{i,k} = (0_{d+d'}, -\mathbb{E}(w_e)_{|_{f_{i,k}}}, 0_{d'})^T$. As shown at center in Figure 1, this attention head yields

$$\tilde{e}_{i,k}^{H2} = (0_{d+d'}, -\mathbb{E}(w_e)_{|_{f_{i,k}}}, 0_{d'})^T.$$

Combining the two attention heads, $\tilde{e}_{i,k}^{H1} + \tilde{e}_{i,k}^{H2}$, and then adding this sum to the skip connection, yields the output from the self-attention layer reflected in Figure 2.

### 3.3. Cross-attention layer for computing $\mathbb{E}(w_e)_{|_{f_{i,k+1}}}$

Within attention block $k + 1 \geq 1$, the output of the self-attention plus skip connection at position $i$ is represented as

$$\tilde{e}_{i,k} = (x_i, w_{e,y_i}, 0_{d'}, f_{i,k+1})^T. \tag{5}$$

We now demonstrate that the expectation

$$\mathbb{E}(w_e)_{|_{f_{i,k}}} = \sum_{c=1}^{C} w_{e,c} \left[ \frac{\exp[w_{e,c}^T f_{i,k}]}{\sum_{c'=1}^{C} \exp[w_{e,c'}^T f_{i,k}]} \right] \quad (6)$$

$$= W_e \cdot \text{softmax}(W_e^T f_{i,k}) \quad (7)$$

may be framed as softmax-based attention, with query $f_{i,k}$, keys $\{w_{e,c}\}_{c=1,C}$ and values $\{w_{e,c}\}_{c=1,C}$. We leverage this fact to update the expectation via a softmax-based cross-attention layer.

Vectors $\{\tilde{e}_{i,k}\}_{i=1,N+1}$ are used to constitute the queries associated with the cross-attention layer, with $W_Q \tilde{e}_{i,k} = f_{i,k+1}$ (this $W_Q$ and related matrices are connected to cross-attention, and are distinct from those discussed above in the context of self attention). The cross-attention matrices $W_K$ and $W_V$ are applied to the $C$ columns of $W_e$, corresponding to the (learned) embedding vectors for the categories/tokens. Specifically, $W_K = W_V$ are both the identity matrix, such that $W_K W_e = W_e$. Using softmax-based attention, one exactly implements (6). The result of cross attention is added to the skip connection, with the updated expectation placed in the position of the aforementioned erasure.

At the output of the last attention block ($K$th, for $K$ GD steps), the vector at position $i$ is

$$e_{i,K} = (x_i, w_{e,y_i}, \mathbb{E}(w_e)_{|_{f_{i,K}}}, f_{i,K})^T . \quad (8)$$

A $d' \times (d + 3d')$ matrix $R$ is used after the last attention block, yielding $f_{i,K} = R e_{i,K}$. At position $i = N+1$, this is then input into the softmax in (1), to predict the probability that $y_{N+1} = c$, for all $c \in \{1, \ldots, C\}$.

### 3.4. Connection to original language Transformer

The original Transformer design (Vaswani et al., 2017) considered a form of interleaved self-attention layers and cross-attention layers (with skip connections) in its decoder. This formed the same type of attention *blocks* we developed above. We have provided some perspective on such a formalism through consideration of inference in terms of functional GD. The difference with that prior work is that here the cross attention is performed with the embedding vectors (columns of $W_e$), where in Vaswani et al. (2017) the cross attention was performed on the encoded input text (prompt).

From another perspective, the expectation as expressed in (7) is evocative of the multi-layered perceptron (MLP) layer employed in current Transformer-based language models (employing only a decoder arm (Brown et al., 2020), and no cross-attention). However, the softmax function is *not* a *pointwise* nonlinearity, as typically employed in MLPs.

### 3.5. Simplification for a single step of GD

As demonstrated when presenting experiments in Section 6.3, Transformer inference based on a single attention block (corresponding to one step of GD) often yields results almost as good as a multi-layered (multi-step GD) model. If

one is only interested in a single GD step, we may simplify the design summarized in Figure 2. Since only one GD step is computed, there is no need to update the expectation $\mathbb{E}(w_e)_{|_{f_{i,0}}} = \bar{w}_e \to \mathbb{E}(w_e)_{|_{f_{i,1}}}$. Consequently, within the self-attention layer we remove the expectation-erasure attention head. Further, since the expectation is not updated, the cross-attention layer is removed entirely. There is only one self-attention layer, using a single attention head. Further, there is no need to isolate the expectation within the encoding, as it is not updated. Therefore, for implementation of a single step of GD, we consider simplified input

$$e_{i,0} = (x_i, w_{e,y_i} - \bar{w}_e, 0_{d'})^T . \quad (9)$$

Details of the parameters for this single-step-GD self-attention layer are presented in Appendix C.

### 3.6. Extensions for modeling language

When presenting results for language modeling from the perspective of functional GD (in Section 6), we will employ positional embedding vectors (Vaswani et al., 2017) as covariates $x_i$. In that connection, concerning the function update in Section 3.2.1, for language modeling we will consider *multiple* self-attention heads for the function update. This may be viewed as extending the discussion of Section 2.2 to $f_\phi(x) = \sum_{m=1}^{M} A_m \psi_m(x)$, where here we assume $M$ attention heads related to the function $f_\phi(x)$, with distinct $\psi_m(x)$ for each. As discussed in Appendix H, in practice we consider $\kappa(\Lambda_m x_i, \Lambda_m x_j)$, where $\Lambda_m$ is a head-dependent *diagonal* matrix (and $\kappa(\cdot, \cdot)$ is the same for all $M$ heads). The same form of $W_V$ as discussed in Section 3.2.1 is used for each of these heads.

In this setup, the erasure self-attention head and the cross-attention design remain unchanged. The multiple kernels for representation of $f_\phi(x)$ also connect to the multi-headed attention used in language models (Vaswani et al., 2017).

## 4. On Learning and Inference

For learning the parameters of the attention-based model, it is assumed that we have access to $L$ contextual datasets, *i.e.*, for $l = 1, \ldots, L$, $\mathcal{C}^{(l)} = \{(x_i^{(l)}, y_i^{(l)})\}_{i=1,N+1}$. When training, we seek to maximize the expected log-likelihood of $p_\phi(Y = y_{N+1}^{(l)} | X = x_{N+1}^{(l)})$, averaging across $l$. For each contextual dataset, it is assumed there is an underlying $f_{\phi^{(l)}}(x) \in \mathcal{F}$ which is to be inferred approximately, based on the associated $N$ contextual examples.

The model parameters include $W_Q$, $W_K$, $W_V$, and $P$ for each attention head at each layer, $W_e$, and $R$ at the Transformer output, prior to the softmax model in (1). We follow terminology from von Oswald et al. (2023): a *Trained TF* is a Transformer for which all model parameters are learned from a random initialization, without any assumptions or restrictions; and a *GD* Transformer is one for which we specify the form of the parameters to be consistent with the above GD-based design. When performing learning for GD,

since the *form* of the parameters is specified, the number of attention parameters is markedly reduced, and $W_e$ is also learned.

## 5. Theoretical Analysis

We analyze the loss-landscape of attention model weights for in-context-learning of categorical data. Consider an $L$-layer attention model, and let $f_{N+1,L} \in \mathbb{R}^{d'}$ denote our estimate of $f(x_{N+1})$ based on the model's output at layer $L$. See Appendix I for details on how $f_{N+1,L}$ is defined as a function of the model's output.

Consider the *population* cross-entropy loss, defined as

$$\bar{\mathcal{L}}(\theta) = \mathbb{E}_{(x_i,y_i)_{i=1...N+1}} \left[ \log \left[ \frac{\exp\left[w_{e,y_{N+1}}^T f_{N+1,L}\right]}{\sum_{c=1}^{C} \exp\left[w_{e,c}^T f_{N+1,L}\right]} \right] \right],$$

where $\theta$ denotes attention model weights (*e.g.*, $W_Q$, $W_K$ and $W_V$ at each layer) and shows up on the RHS through $f_{N+1,L}$. $\bar{\mathcal{L}}$ is the loss from our prediction of the query token ($y_{N+1}$), in expectation over possible realizations of the context $(x_i, y_i)_{i=1...N+1}$. In the limit of infinite training data, the model weights $\theta$, trained by a gradient-based optimization algorithm, will converge to a **stationary point** of $\bar{\mathcal{L}}$. In practice, as discussed in Section 4, the expectation $\mathbb{E}_{(x_i,y_i)_{i=1...N+1}}$ associated with $\bar{\mathcal{L}}(\theta)$ is estimated with $L$ contextual training sets.

**Theorem 1 (Informal Statement of Theorem 2):**
*Assume that (1) the self-attention only involves the feature vectors $x_i$'s, (2) the covariate vectors $x_i$'s have distribution that is rotationally invariant (3) the cross-attention layer as described in Section 3.3 exactly computes $\mathbb{E}(w_e)_{|_{f_{i,k}}}$. For $i = 1...N+1$, let $f_{i,k}$ denote the readout of the model's prediction for $f(x_i)$ at layer $k$. Then there exists a choice of attention-model parameters $\hat{\theta}$, which **exactly implement gradient descent**, i.e. $f_{i,k}$ evolves according to (2)-(3). Furthermore, $\hat{\theta}$ is a **stationary point** of $\bar{\mathcal{L}}(\theta)$.*

We present the formal version of Theorem 1 as Theorem 2 in Appendix I. We highlight aspects of Theorem 1 below:

**1. Convergence to GD Implementation:** As the number of training samples increases, the empirical loss over the training set approaches the population loss $\bar{\mathcal{L}}$. Consequently, Theorem 1 suggests that the $K$-layer model, after training converges, will implement $K$ steps of functional GD. Theorem 1 is supported by Figure 3, which suggests Trained TF converges in accuracy to GD with more training samples. Additionally, Figure 11 in Appendix F shows that the attention parameter matrices at training convergence do coincide with the stationary point parameters constructed in Appendix I. Figure 5(left) shows a similar match between GD and Trained TF for two layers.

**2. Challenges of Categorical Data:** Previous analysis of the Transformer optimization landscape have focused on the

regression setting, where the observations $y_i$ are equal to the target function $f(x_i)$ (up to a simple Gaussian perturbation). Their (Cheng et al., 2024) proofs do not generalize to the categorical setting, where the observations $y_i$ are **sampled** conditional on $f(x_i)$ as in (1). Importantly, $f(\cdot)$ is a **multi-dimensional unobserved latent function.** The categorical setting is **crucial for modelling language.**

**3. Importance of Softmax:** Softmax in cross-attention enables the Transformer to compute $\mathbb{E}(w_e)_{|_{f_{i,k}}}$, as detailed in Section 3.3, which is crucial for implementing GD for categorical data, and plays a key role in the proof of Theorem 1. Softmax-activated self-attention generalizes the kernel-based attention considered in Cheng et al. (2024).

## 6. Experiments

All experiments were performed on a Tesla V100 PCIe 16 GB GPU. Code needed to replicate our experiments is at `https://github.com/aarontwang/icl_attention_categorical`.

### 6.1. Synthetic data

We consider synthetic data generated by the model $p(Y = c | f(x)) = \exp[w_c^T f(x)] / \sum_{c'=1}^{C} \exp[w_{c'}^T f(x)]$, for $C = 25$ and $w_c \in \mathbb{R}^5$. The data are designed such that $f(x)$ is highly nonlinear and changes for each set of contextual data. Details on data generation are provided in Appendix E.

We compare test prediction performance of attention-based models, based on Trained TF and GD. In all of these experiments, $N = 10$ data pairs $\mathcal{C}^{(l)} = \{(x_i^{(l)}, y_i^{(l)})\}_{i=1,N}$ define the context, with an associated query pair $(x_{N+1}^{(l)}, y_{N+1}^{(l)})$ ($y_{N+1}^{(l)}$ is known when training, not when testing). We train the model with contexts $\mathcal{C}^{(l)}$ for $l = 1, \ldots, L$, and various training sizes $L$ are considered. Testing performance is performed using a *distinct* set of contextual data and query, with data generated as above, with performance averaged over contexts $\mathcal{C}^{(L+m)}$, for test sets $m = 1, \ldots, M$ ($M = 2048$), each with an associated query $x_{N+1}^{(L+m)}$.

**For a single self-attention layer, there is close agreement between GD and Trained TF** (Figure 3). From the theory of Section 5, we expect the Trained TF and GD to yield similar results. Considering the average Top-1 model predictive accuracy and the average negative log-likelihood (on the test data), we observe close agreement between Trained TF and GD after a sufficiently large number $L$ of in-context training sets. The GD-based self-attention model requires much smaller $L$ to train well. We also examined linear attention, for which $\kappa(x_i, x_j) = x_i^T x_j$; as expected, it performed poorly because the underlying $f_\phi(x)$ is highly nonlinear.

Figure 3 shows close agreement in the *predictive performance* of the Trained TF and GD, but that doesn't necessarily mean that the underlying parameters learned by Trained

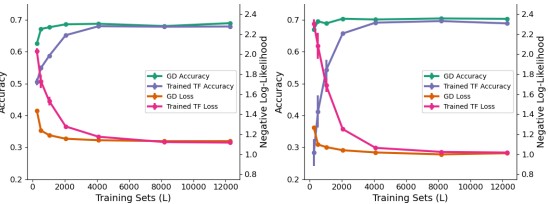

*Figure 3.* For one self-attention layer, comparison of accuracy and loss of GD and Trained TF with RBF (left) and softmax (right) attention, as a function of contextual training sets. Error bars are computed from five different random initializations.

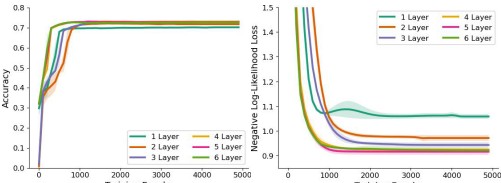

*Figure 4.* For the multi-layered model in Section 3, training curves of accuracy (left) and negative log-likelihood (right) for the GD transformer with softmax attention. Error bars are computed from five different random initializations.

TF agree with the GD theory. In Appendix F, we consider another synthetic-data example of lower dimension (for visualization), and there we show close agreement between the Trained TF learned parameters and those of the GD framework, as predicted by the analysis in Section 5.

**Additional attention layers improve GD-attention predictive accuracy.** In Figure 4 we consider the same data as discussed above, and consider the GD-based attention model for up to six attention blocks (corresponding to up to six GD steps). In these experiments, we trained with $L = 2048$ contextual blocks, and performance is shown *on the test data* as a function of number of training epochs. We note that a single attention layer achieves good Top-1 predictive accuracy, but a noticeable improvement in predictive accuracy is observed by adding a second attention block. The improvement in the (test data) negative log-likelihood is more substantial with increasing attention-block layers.

**With sufficient training data for Trained TF, there is close agreement between Trained TF and GD for two attention blocks** (Figure 5(left)). However, note that a large number of contextual data (*e.g.*, $L > 25,000$) are needed to train this multi-layered Trained TF model, while far less training data (*e.g.*, $L < 5000$) are needed for the GD model. Comparing with the single-layer model in Figure 3, note that far more training data is needed for Trained TF when going from a one-layer model to a two-layer model.

### 6.2. In-Context Image Classification

Using the ImageNet dataset (Russakovsky et al., 2014), we select 900 classes for Transformer training, and a separate 100 classes for testing. For each contextual set $\mathcal{C}^{(l)}$, 5 distinct classes are selected uniformly at random, and for each

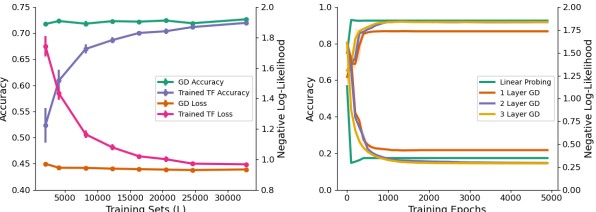

*Figure 5.* (Left) For two attention blocks for the model in Section 3, comparison of accuracy and loss of GD and Trained TF with softmax attention, as a function of contextual training sets. (Right) Model accuracy for the ImageNet dataset, for which the attention-based model was trained once for 500 epochs, and the linear probing was trained for 500 epochs on each test context.

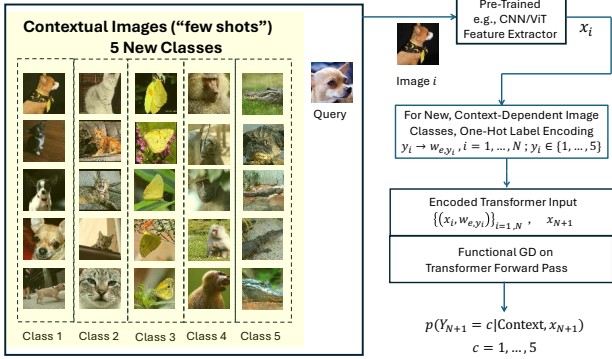

*Figure 6.* Illustration of in-context learning with image data. Each image is mapped to a feature vector $x_i$. Contextual data are composed of a set of images for new classes (here five), and a query image that, in the Transformer forward pass, is assigned a probability of being associated with each of the contextual labels.

such class 10 specific images are selected at random, and therefore $N = 50$ (image $N + 1$ is selected at random from the 5 class types considered in the context data). When training $L = 2048$, and test performance is averaged for $M = 2048$. The covariates $x_i$ (with $d = 512$) correspond to features from the VGG network (Simonyan & Zisserman, 2015). We set $d' = 4$ and learn $\{w_c\}_{c=1,C}$. See Figure 6 for a depiction of ICL for image classification.

**For in-context ImageNet classification, GD-based attention yields Top-1 predictive accuracy comparable to "linear probing"** (Figure 5 (right)). Importantly, with linear probing a model *must be learned anew* for each test contextual block, to be contrasted with the attention-based model, for which no fine-tuning of parameters are done after training. We observe that with one layer of GD with self-attention, the GD attention model achieves slightly lower accuracy and higher loss compared to the linear probing model. However, with two and three attention *blocks*, the performance of the GD transformer aligns almost exactly with the linear probing model. Note that with the VGG model for feature extraction, and our in-context model for classification, new image-based contextual data (with labels never seen before) can be classified with no parameter

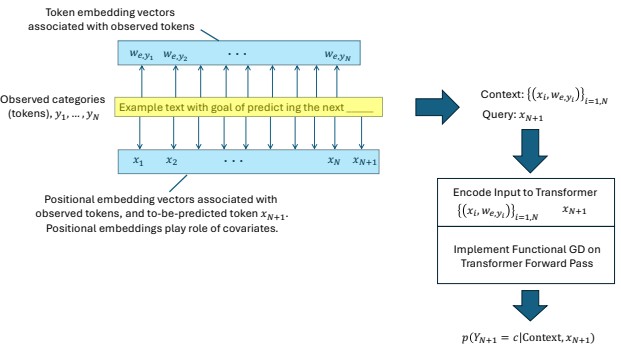

*Figure 7.* Summary of how language modeling is performed with ICL. Token $y_i$ is mapped to a learned embedding vector $w_{e,y_i}$, and positional embeddings are used as covariates $x_i$. The input text constitutes the sequence of $N$ tokens, and the next token is connected to the query at position $x_{N+1}$.

adjustment (fine tuning).

### 6.3. Language Modeling

We examine language generation in the context of GD-based attention models. We make comparisons to text generated by a Transformer architecture, following the design of (Radford et al., 2019). For these experiments, we train on a combination of the *Tiny Stories* (Eldan & Li, 2023) and *Children Stories*[1].

We have performed experiments for multi-layered models. However, it has been shown that for such simple stories a single-layer model often generates good text (Eldan & Li, 2023) (and the experiments from the previous section on categorical classification showed that a one-layer model yields good "first-order" category/token prediction). For simplicity, in the language-generation experiments we consider a single-layer GD model and a single-layer Transformer.

For both the GD and Transformer based models, embedding vectors are learned for each token, with $C = 50,257$ *unique* tokens represented and an embedding dimension $d' = 512$; 8 attention heads are use for both models. Additionally, positional embedding vectors are learned for each of the 256 positions in our model's context window, with an additional 257th position learned for the GD model (for position $x_{N+1}$). We define $p_i$ as the positional embedding at position $i$ and (as before) $w_{e,y_i}$ as the corresponding token embedding for a given sequence ($y_i \in \{1, \ldots, C\}$ is the token index at position $i$).

Concerning the GD-based attention model for language (summarized in Figure 7), recall that covariates $x_i$ are assumed for each position, *including the query* $x_{N+1}$. To ensure we are able to provide $x_{N+1}$ for a context of size $N$, we use exclusively positional embeddings for our covariates, as we can easily learn $p_{N+1}$ (referenced above). The

[1]https://huggingface.co/datasets/ajibawa-2023/Children-Stories-Collection

values only correspond to indices $i = 1, \ldots, N$ for which we have tokens, and therefore the values are set to $w_{e,y_i}$. For predicting the token at position $N + 1$, $p_{N+1}$ is used for the query. Hence, the covariates $x_i$ correspond to $p_i$, and attention is based entirely on token position. This is an important departure from the Transformer model, which performs attention using $w_{e,y_i} + p_i$ for its queries, keys, and values, and makes a prediction based on the output at position $N$. Consistent with GD only using $w_{e,y_i}$ for the values, we note that many modern language models, such as LLaMa, also do not incorporate positional embeddings in their value vectors (Touvron et al., 2023).

Another key difference between the GD and Transformer is that the latter effectively does not initialize the underlying function as the zero vector (there is no scratch space with zeros), where the GD-based initializes the latent function $f_{i,0} = 0_{d'}$. Despite these limitations of GD, we show below it performs relatively well compared to the Transformer.

In testing both the GD-based model and the Transformer, the model is tasked with generating the ending to a *previously unseen* story. To demonstrate the range in performance of each model, we show examples of both a "good" generated ending and a "bad" generated ending (for both models, about 50% of the examples were of each type). We have done extensive experimentation; many other examples are presented in Appendix K.

**The generated text from the GD-based language model are of similar qualitative character as a comparable Transformer-based model**. In Table 1 we present GD generated text based on *softmax* attention, and in Table 2 are shown results from the Transformer (we choose the softmax variant of GD because it aligns better with the Transformer's attention; results for other attention kernels are presented in Appendix K). For both the GD and Transformer, the failure model of the bad examples is often repetition. The high quality of the simple GD-based "good" examples is deemed remarkable, given the relative simplicity of this model. In addition to the aforementioned limitations of the GD-based model, the Transformer employs a feedforward element after the attention and skip connection, while GD does not. For consistency, we included layernorm (Xiong et al., 2020) at the output of both models, before the softmax calculation.

Recall that the purpose of this analysis is *not* to suggest that a GD-based model can replace the Transformer. Rather, it is believed that studying GD-based text generation may given insights into the Transformer, possibly suggesting ways it can be improved or simplified.

**A quantitative analyis based on GPT-4o shows that the GD-based language model can achieve similar quantitative performance to a comparable Transformer, with far fewer parameters.** A similar GPT-4 based quantitative

### Softmax GD Model (Good Example)

"Once upon a time, in a small town named Harmonyville, lived two best friends, Max and Lily. Even though they were far away from each other due to some circumstances, they always found innovative ways to keep their bond strong. Max had moved to another city for his studies while Lily was still living in Harmonyville. One sunny day, Max discovered how social media could make communication easier between them. He learned that messages sent over social media traveled almost instantly across vast distances! These electronic signals zipped through wires and airwaves at incredible speeds, allowing Max and Lily to talk in real-time. It felt like they were transported into a world full of energy! Lily and Max couldn't wait to share their newfound knowledge with others. However, they soon realized something unexpected happened. A storm hit Harmonyville, causing chaos throughout the town. Everyone was shocked and worried. They didn't know what to do. Lily and Max decided to talk to Dr. Maple, a wise old scientist known for her love of science and technology. She explained that when people feel overwhelmed, they might feel "

### Softmax GD Model (Bad Example)

"Once there was a happy dog named Spot. Spot liked to run and play, but he was too big. His weight was so heavy that he couldn't run around like he wanted to. Spot hated his weight. He wished he could be as light as the wind and be able to run freely. But no matter what Spot did, he couldn't seem to change. One day, Spot went on an adventure and found a clear pond. He looked into the water and saw his reflection. Spot hated the way he looked. He wanted to see what he could do. He said, "Let's go to the pond and see if he could find it. He was so happy that he could help his friends. The moral of the story is to always be kind to help others. The moral of the story is to always be kind to help others. The moral of the story is to always be kind to help others. The moral of the story is to always be kind to help others. The moral "

*Table 1.* Generations from the GD model with softmax attention. The black font represents the prompt, and the red font represented generated language.

### Single-Layer Transformer (Good Example)

"Once upon a time, there was a big snake named Sam. Sam was a flexible snake who could bend and twist in many ways. He lived in a forest with many friends. One day, Sam met a little bird named Betty. Betty was sad because she lost her to y. Sam said, "Don't worry, Betty, I will help you find your toy!" They looked for the toy together. They looked for a minute, then two, then three. Finally, they found the toy under a big tree. Sam was so happy! He said, "Thank you, Bett y! You are a good friend." From that day on, Sam and Betty became best friends. They played together every day. Sam learned that helping each other is more important than having fun. And they lived happily ever after. The moral of the story is to always help your friends when they need it. Sam and Betty learned that it is important to be kind "

### Single-Layer Transformer (Bad Example)

"Once upon a time, there was a yellow swan. The swan lived in a small pond. The pond had pretty flowers and green grass all around it. One day, the swan met a big frog. The frog was not nice. The frog said, "I want to play a game." The swan did not want to play, but the frog was very big. The swan was scared. The frog suggest a game to see who can swim faster. The swan swam very fast. The swan swam very fast. The swan swam very fast. The swan swam and swam until it reached the pond. The swan was so happy to see the swan again. From that day on, the swan and the swan became best friends. They swam and played together every day. The swan and the swan became best friends. They swam and swam together every day "

*Table 2.* Generations from the single-layer Transformer model. The black font represents the prompt, and the red font represented generated language.

analysis was considered in Eldan & Li (2023) (more implementation details in Appendix K). Each model was tasked with generating the ending to a story, as above, but now these responses were graded by OpenAI's GPT-4o model. The grading model was told to evaluate the quality of the generated text based on the start of the story (see Appendix J for details). It then provided a score out of 10 for the generation's grammer, its consistency with the start of the story, its general plot, and it's creativity. The table shows the average scores across 200 story generations.

| Model | Grammar | Consistency | Plot | Creativity | Overall |
|---|---|---|---|---|---|
| True Ending | 9.21 | 8.77 | 7.77 | 6.68 | 8.11 |
| Softmax GD | 4.25 | 3.40 | 3.08 | 3.76 | 3.62 |
| Softmax GD (FF) | 5.90 | 4.55 | 4.22 | 4.70 | 4.84 |
| Transformer | 5.97 | 4.98 | 4.43 | 4.75 | 5.04 |

*Table 3.* GPT-4o scoring of the generated story endings. Each item is graded out of a maximum score of 10. Softmax GD and Softmax GD with FF have 8K attention-weight parameters, and the Transformer has 6M attention-weight parameters.

The overall scoring of both the 1-layer GD and Transformer models were low, relative to the true ending. However, while the Transformer yields better scores than GD, these scores are much closer to each other. Importantly, the GD model is able to capture much of the generation power of the Transformer model. To examine the importance of the feedforward (FF) element (plus associated skip connection) in the Transformer, which is absent from a GD perspective, we kept all aspects of the GD-based model unchanged, but appended a FF network above the attention output from position $N + 1$, before the associated output goes to softmax over tokens. Note from Table 3, *the GD formulation plus an added FF element yields performance almost the same as the Transformer*! Associated examples of generated text are shown in Appendix K. This is deemed an important insight into the relationship between the GD perspective and the Transformer, and the importance of the FF element to Transformer performance. The performance of GD-based attention plus FF relative to the Transformer is particularly noteworthy, given that the GD-based model has far fewer parameters (see the caption in Table 3).

## 7. Conclusions

Through careful analysis and novel theory, as well as detailed experiments, including on two real datasets, we have investigated in-context learning with categorical observations. This analysis has led to a new attention-block framework that can perform multi-step functional GD exactly. Our language-generation experiments indicate that a GD-based attention-model perspective, with the addition of an appended feedforward (FF) element, can generate text (trained on a particular corpus) as well as a comparable Transformer, but with far fewer parameters. More work is needed to investigate how the FF element, coupled with the GD-based attention model, works with our new attention-based model to yield such high-quality results.

## Acknowledgments

The work reported here was supported by the Office of Naval Research under grant 313000130. We also thank the anonymous reviewers for constructive suggestions, that have impacted the final version of the paper.

## Impact Statement

This paper presents work whose goal is to advance the field of Machine Learning, particularly in the context of language modeling. There are many potential societal consequences of our work, particularly in terms of enhanced understanding of the mechanisms that underlie language models. Insights from this line of work may lead to better and/or more efficient language models in the future. This could, for example, reduce energy requirements for such systems, via enhance efficiency.

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

# APPENDIX

## A. Summary of Content in this Appendix

This Appendix provides an extensive set of details connected to material in the paper, as well as additional experiments. To aid the reader in navigating and using this Appendix, we summarize what is provided, and in what section (with a hyperlink to it).

- **Section B**: Derivation of the GD Update Equation

- **Section C**: Transformer Parameters for Single GD Step

- **Section D**: Transformer Parameters for Exact Multi-Step GD with Interleaved Attention Layers

- **Section E**: Details on Generation of the Synthetic Data in the Main Paper

- **Section F**: Lower-Dimensional Dataset for Further Understanding of Trained TF and Connection to GD

- **Section G**: Details on the Gradient Updates for Real $Y$ and Gaussian $p(Y|X = x)$

- **Section H**: Multiple Kernels for Function Modeling

- **Section I**: Proof of Identity Stationary Point

- **Section K**: Additional Language-Model Experiments

## B. Derivation of the GD Update Equation

The cost function for inferring the parameters $\phi = (A, b)$, with $A \in \mathbb{R}^{d' \times m}$ and $b \in \mathbb{R}^{d'}$, may be expressed as

$$
\mathcal{L}(A, b) = -\frac{1}{N} \sum_{i=1}^{N} \log \left[ \frac{\exp[w_{e,y_i}^T (A\psi(x_i) + b)]}{\sum_{c=1}^{C} \exp[w_{e,c}^T (A\psi(x_i) + b)]} \right]
$$

$$
= -\frac{1}{N} \sum_{i=1}^{N} [w_{e,y_i}^T A\psi(x_i) + w_{e,y_i}^T b
$$

$$
- \log \sum_{c=1}^{C} \exp(w_{e,c}^T A\psi(x_i) + w_{e,c}^T b)] . \quad (10)
$$

Taking the partial derivative of $\mathcal{L}$ wrt $b_j$, component $j$ of $b$:

$$
\frac{\partial}{\partial b_j} \mathcal{L} = -\frac{1}{N} \sum_{i=1}^{N} [w_{e,y_i}(j)
$$

$$
- \frac{\sum_{c=1}^{C} \exp[w_{e,c}^T (A\psi(x_i) + b)] w_{e,c}(j)}{\sum_{c'=0}^{C} \exp[w_{e,c'}^T (A\psi(x_i) + b)]} ]
$$

$$
= -\frac{1}{N} \sum_{i=1}^{N} [w_{e,y_i}(j) - \frac{\sum_{c=1}^{C} \exp[w_{e,c}^T f_i] w_{e,c}(j)}{\sum_{c'=0}^{C} \exp[w_{e,c'}^T f_i]}] ,
$$

where $w_{e,y_i}(j)$ is component $j$ of $w_{e,y_i} \in \mathbb{R}^{d'}$, and $f_i = A\psi(x_i) + b$. Therefore

$$
\nabla_b \mathcal{L} = -\frac{1}{N} \sum_{i=1}^{N} \left[ w_{e,y_i} - \frac{\sum_{c=1}^{C} \exp[w_{e,c}^T f_i] w_{e,c}}{\sum_{c'=0}^{C} \exp[w_{e,c'}^T f_i]} \right]
$$

$$
= -\frac{1}{N} \sum_{i=1}^{N} \left[ w_{e,y_i} - \mathbb{E}(w_e)_{|f_i} \right] . \quad (11)
$$

We consequently have the GD update rule for $b$

$$
b_{k+1} = b_k + \frac{\alpha}{N} \sum_{i=1}^{N} \left[ w_{e,y_i} - \mathbb{E}(w_e)_{|f_{i,k}} \right] . \quad (12)
$$

Similarly, let $a_j$ represent the $j$th row of $A$. Taking the gradient of $\mathcal{L}$ wrt $a_j$:

$$
\nabla_{a_j} \mathcal{L} = -\frac{1}{N} \sum_{i=1}^{N} \left[ w_{e,y_i}(j)\psi(x_i) \right.
$$

$$
\left. - \frac{\sum_{c=1}^{C} \exp[w_{e,c}^T (A\psi(x_i) + b)] w_{e,c}(j)\psi(x_i)}{\sum_{c'=0}^{C} \exp(w_{e,c'}^T (A\psi(x_i) + b))} \right]
$$

$$
= -\frac{1}{N} \sum_{i=1}^{N} \left[ w_{e,y_i}(j) \right.
$$

$$
\left. - \frac{\sum_{c=1}^{C} \exp[w_{e,c}^T f_i] w_{e,c}(j)}{\sum_{c'=0}^{C} \exp(w_{e,c'}^T f_i)} \right] \psi(x_i)
$$

$$
= -\frac{1}{N} \sum_{i=1}^{N} \left[ w_{e,y_i}(j) - \mathbb{E}(w_e(j))_{|f_i} \right] \psi(x_i) . \quad (13)
$$

The gradient update step for $a_j$ is

$$
a_{j,k+1} = a_{j,k} - \alpha \nabla_{a_j} \mathcal{L}
$$

$$
= a_{j,k} + \frac{\alpha}{N} \sum_{i=1}^{N} \left[ w_{e,y_i}(j) - \mathbb{E}(w_e(j))_{|f_i} \right] \psi(x_i) .
$$

Using the GD update rules for $b$ and $\{a_j\}_{j=1,d'}$, we have

$$
f_{j,k+1} = \begin{pmatrix} a_{1,k+1}^T \psi(x_j) + b_{1,k+1} \\ \vdots \\ a_{d',k+1}^T \psi(x_j) + b_{d',k+1} \end{pmatrix}
$$

$$
= f_{j,k} + \frac{\alpha}{N} \sum_{i=1}^{N} \left[ w_{e,y_i} - \mathbb{E}(w_e)_{|f_{i,k}} \right] \kappa(x_i, x_j)
$$

$$
+ \alpha \sum_{i=1}^{N} \left[ w_{e,y_i} - \mathbb{E}(w_e)_{|f_{i,k}} \right] . \quad (14)
$$

## C. Transformer Parameters for Single GD Step

The GD update equation for the underlying function $f_\phi(x)$ is repeated here for convenience:

$$f_{j,k+1} = f_{j,k} + \underbrace{\frac{\alpha}{N} \sum_{i=1}^{N} [w_{e,y_i} - \mathbb{E}(w_e)_{|f_{i,k}}] \kappa(x_i, x_j)}_{\Delta f_{j,k}} .$$

$$(15)$$

To effect a realization of the Transformer that implements this update rule in its forward pass, we consider the first attention layer, which we can generalized to an arbitrary number of layers. The desired input vectors at the first layer are

$$e_{i,0} = \begin{pmatrix} x_i \\ w_{e,y_i} - \mathbb{E}(w_e)_{|f_{i,0}=0_{d'}} \\ 0_{d'} \end{pmatrix}$$

$$= \begin{pmatrix} x_i \\ w_{e,y_i} - \frac{1}{C}\sum_{c=1}^{C} w_{e,c} \\ 0_{d'} \end{pmatrix} . \quad (16)$$

We have initialized $f_{i,0} = 0_{d'}, \; \forall\, i \in \{1, \ldots, N+1\}$, which corresponds to the last $d'$ elements of $e_{i,0}$. The input for query $N+1$ is

$$e_{N+1,0} = \begin{pmatrix} x_{N+1} \\ 0_{d'} \\ 0_{d'} \end{pmatrix} . \quad (17)$$

Note that $y_{N+1}$ is not known the Transformer at input.

The category embedding vectors reflected in the columns of $W_e$ are to be learned. They are used in the input encoding, as discussed above, and also within the softmax over categories at the Transformer output. Motivated by the need to learn $W_e$, in practice we encode the input vectors as:

$$e_{i,0} = \begin{pmatrix} x_i \\ \tilde{y}_i - \frac{1}{C} 1_C \\ 0_{d'} \end{pmatrix} , \quad (18)$$

where $\tilde{y}_i$ is a one-hot encoding of category $y_i$; *i.e.*, $\tilde{y}_i$ is a vector of all zeros, except a single 1, at position $y_i + 1$, with $y_i \in \{0, \ldots, C\}$, and $1_C$ is a $C$-dimensional vector of all ones. The input for query $N+1$ (for which we do not have $y_{N+1}$ as Transformer input) is

$$e_{N+1,0} = \begin{pmatrix} x_{N+1} \\ 0_{d'} \\ 0_{d'} \end{pmatrix} . \quad (19)$$

The first step at the Transformer input, prior to the first attention layer for implementation of GD, is to multiply $e_{i,0}$ by the matrix

$$\begin{pmatrix} I_{d \times d} & 0_{d \times C} & 0_{d \times d'} \\ 0_{d' \times d} & W_e & 0_{d' \times d'} \\ 0_{d' \times d} & 0_{d' \times C} & I_{d' \times d'} \end{pmatrix} , \quad (20)$$

where $W_e \in \mathbb{R}^{d' \times C}$. This implements the initial encoding of the vectors for implementing GD, and it separates out the matrix $W_e$ for learning.

After this initialization, the vectors are then input to the attention layer, which implements computation of $\Delta f_{i,0}$ for all positions $i$, with keys and values on inputs $i = 1, \ldots, N$ and queries for all $j = 1, \ldots, N+1$. To implement this, we design the query and key matrices as follows

$$W_Q = W_K = \begin{pmatrix} I_{d \times d} & 0_{d \times d'} & 0_{d \times d'} \\ 0_{d' \times d} & 0_{d' \times d'} & 0_{d' \times d'} \\ 0_{d' \times d} & 0_{d' \times d'} & 0_{d' \times d'} \end{pmatrix} , \quad (21)$$

such that $W_Q e_{i,0} = (x_i, 0_{2d'})^T$, and then these are used in the kernel attention as $\kappa(W_Q e_{j,0}, W_K e_{i,0})$.

For the values, we need to extract $\frac{\alpha}{N}[w_{e,y_i} - \mathbb{E}(w_e)_{f_{i,0}}]$ from the input vectors, so the value matrix is

$$W_V = \frac{\alpha}{N} \begin{pmatrix} 0_{d \times d} & 0_{d \times d'} & 0_{d \times d'} \\ 0_{d' \times d} & I_{d' \times d'} & 0_{d' \times d'} \\ 0_{d' \times d} & 0_{d' \times d'} & 0_{d' \times d'} \end{pmatrix} , \quad (22)$$

such that $W_V e_{i,0} = (0_d, \frac{\alpha}{N}[w_{e,y_i} - \frac{1}{C}\sum_{c=1}^{C} w_{e,c}], 0_{d'})^T$. This goes into the attention network, the output of which is $(0_d, \Delta f_{j,k}, 0_{d'})^T$ at each position $j$.

Finally, we multiply the output of $W_V e_{i,0}$ by a matrix $P$, which corresponds to

$$P = \begin{pmatrix} 0_{d \times d} & 0_{d \times d'} & 0_{d \times d'} \\ 0_{d' \times d} & 0_{d' \times d'} & 0_{d' \times d'} \\ 0_{d' \times d} & I_{d' \times d'} & 0_{d' \times d'} \end{pmatrix} , \quad (23)$$

This output is then added to the input, constituting a skip connection, from which $f_{i,0}$ is updated to $f_{i,1}$.

When we perform model learning for the GD-based form of the Transformer, based on $\{\mathcal{C}^{(l)}\}_{l=1,\ldots,L}$ we use the cross-entropy loss based on the softmax likelihood function. The parameters to be learned are $W_e \in \mathbb{R}^{d' \times C}$ and the learning rate $\alpha$ (plus the kernel parameter). At the output of the last attention layer, the output at position $N+1$ is multiplied by the matrix $\begin{pmatrix} 0_{d' \times d} & 0_{d' \times d'} & I_{d' \times d'} \end{pmatrix}$ which yields an approximation for $f_{\phi_K}(x_{N+1})$ for $K$ layers. This is employed with $W_e$ to manifest softmax probabilities on categories (and cross-entropy loss).

## D. Transformer Parameters for Exact Multi-Step GD with Interleaved Attention Layers

The input to the Transformer at layer $k$ is $e_{i,k} = (x_i, w_{e,y_i}, \mathbb{E}(w_e)_{|f_{i,k}}, f_{i,k})^T$, as summarized in Figures 1 and 2 of the main paper. Each attention block consists of a self-attention layer, composed of two attention heads; one of these attention heads implements $f_{i,k} \to f_{i,k+1}$ like above (for which $\mathbb{E}(w_e)_{|f_{i,k}}$ is needed), and the second attention head erases $\mathbb{E}(w_e)_{|f_{i,k}}$, preparing for its update by the subsequent cross-attention layer.

## D.1. Self-attention layer

In matrix form, the input at layer $k$ is

$$\begin{pmatrix} x_1 & \dots & x_N & x_{N+1} \\ w_{e,y_1} & \dots & w_{e,y_N} & 0_{d'} \\ \mathbb{E}(w_e)_{|f_{1,k}} & \dots & \mathbb{E}(w_e)_{|f_{N,k}} & \mathbb{E}(w_e)_{|f_{N+1,k}} \\ f_{1,k} & \dots & f_{N,k} & f_{N+1,k} \end{pmatrix} \quad (24)$$

The update equation for $f_{i,k+1}$ is given by

$$f_{i,k+1} = f_{i,k} + \Delta f_{i,k} \quad (25)$$

where

$$\Delta f_{i,k} = \frac{\alpha}{N} \sum_{i=1}^{N} (w_{e,y_i} - \mathbb{E}(w_e)_{|f_{i,k}}) \kappa(x_i, x_j) \quad (26)$$

### D.1.1. SELF-ATTENTION HEAD 1

We design $W_K^{(1)}$, $W_Q^{(1)}$, and $W_V^{(1)}$ such that

$$W_K^{(1)} e_{i,k} = (x_i, 0_{d'}, 0_{d'}, 0_{d'})^T \quad (27)$$

$$W_Q^{(1)} e_{j,k} = (x_j, 0_{d'}, 0_{d'}, 0_{d'})^T \quad (28)$$

$$W_V^{(1)} e_{i,k} = (0_d, 0_{d'}, 0_{d'}, \frac{\alpha}{N}[w_{e,y_i} - \mathbb{E}(w_e)_{|f_{i,k}}])^T \quad (29)$$

The output of this first attention head, at position $j \in \{1, \dots, N+1\}$ is

$$(0_d, 0_{d'}, 0_{d'}, \frac{\alpha}{N} \sum_{i=1}^{N} (w_{e,y_i} - \mathbb{E}(w_e)_{|f_{i,k}}) \kappa(x_i, x_j))^T \quad (30)$$

The output of this first attention head at this first attention layer (before adding the skip connection) is

$$O^{(1)} = \begin{pmatrix} 0_d & \dots & 0_d & 0_d \\ 0_{d'} & \dots & 0_{d'} & 0_{d'} \\ 0_{d'} & \dots & 0_{d'} & 0_{d'} \\ \Delta f_{1,k} & \dots & \Delta f_{N,k} & \Delta f_{N+1,k} \end{pmatrix} \quad (31)$$

### D.1.2. SELF-ATTENTION HEAD 2

With the second attention head we want to add $(0_d, 0_{d'}, -\mathbb{E}(w_e)_{|f_{j,k}}, 0_{d'})^T$ from position $j$, so we clear out the prior expectation. This will provide "scratch space" into which, with the next attention layer type, we will update the expectation, using $f_{j,k+1}$. To do this, we design $W_Q^{(2)}$ and $W_K^{(2)}$ such that

$$W_K^{(2)} e_{i,k} = \lambda(x_i, 0_{d'}, 0_{d'}, 0_{d'})^T \quad (32)$$

$$W_Q^{(2)} e_{j,k} = \lambda(x_j, 0_{d'}, 0_{d'}, 0_{d'})^T \quad (33)$$

where $\lambda \gg 1$. With an RBF kernel, for example (similar things will happen with softmax), if $\lambda$ is very large,

$$\kappa(W_K^{(2)} e_{i,k}, W_Q^{(2)} e_{j,k}) = \delta_{i,j} \quad (34)$$

where $\delta_{i,j} = 1$ if $i = j$, and it's zero otherwise.

The value matrix is designed as

$$W_V^{(2)} e_{i,k} = (0_d, 0_{d'}, \mathbb{E}(w_e)_{|f_{i,k}}, 0_{d'})^T \quad (35)$$

The output of this head is

$$O^{(2)} = \begin{pmatrix} 0_d & \dots & 0_d & 0_d \\ 0_{d'} & \dots & 0_{d'} & 0_{d'} \\ \mathbb{E}(w_e)_{|f_{1,k}} & \dots & \mathbb{E}(w_e)_{|f_{N,k}} & \mathbb{E}(w_e)_{|f_{N+1,k}} \\ 0_{d'} & \dots & 0_{d'} & 0_{d'} \end{pmatrix} \quad (36)$$

We then add $P^{(1)}O^{(1)} + P^{(2)}O^{(2)}$, with $P^{(1)}$ and $P^{(2)}$ designed so as to yield the cumulative output of the attention

$$O^{(\text{total})} = \begin{pmatrix} 0_d & \dots & 0_d & 0_d \\ 0_{d'} & \dots & 0_{d'} & 0_{d'} \\ -\mathbb{E}(w_e)_{|f_{1,k}} & \dots & -\mathbb{E}(w_e)_{|f_{N,k}} & -\mathbb{E}(w_e)_{|f_{N+1,k}} \\ \Delta f_{1,k} & \dots & \Delta f_{N,k} & \Delta f_{N+1,k} \end{pmatrix} \quad (37)$$

This is now added to the skip connection, yielding the total output of this attention layer as

$$T = \begin{pmatrix} x_1 & \dots & x_N & x_{N+1} \\ w_{e,y_1} & \dots & w_{e,y_N} & 0_{d'} \\ 0_{d'} & \dots & 0_{d'} & 0_{d'} \\ f_{1,k+1} & \dots & f_{N,k+1} & f_{N+1,k+1} \end{pmatrix} \quad (38)$$

With the first attention layer, with two heads, we update the functions, and we also erase the prior expectations. In the next attention layer, we update the expectations, and place them in the locations of the prior expectations.

## D.2. Cross-Attention Layer

The vectors connected to $T$ above will go into the next attention layer, where we wish to update

$$\mathbb{E}(w_e)_{|f_{i,k+1}} = \sum_{c=1}^{C} w_{e,c} \exp\left[\frac{w_{e,c}^T f_{i,k+1}}{\sum_{c'=1}^{C} w_{e,c'}^T f_{i,k+1}}\right] \quad (39)$$

in the aforementioned scratch space.

In this setup, the keys and values will be defined in terms of

$$W_e = \left( w_{e,0}, \ldots, w_{e,c} \right) \qquad (40)$$

There are many ways this can be done. Consider

$$W_K w_{e,c} = \begin{pmatrix} 0_d \\ 0_{d'} \\ 0_{d'} \\ w_{e,c} \end{pmatrix} \qquad (41)$$

$$W_V w_{e,c} = \begin{pmatrix} 0_d \\ 0_{d'} \\ w_{e,c} \\ 0_{d'} \end{pmatrix} \qquad (42)$$

Let $T_i$ represent the $i$th column of $T$, then

$$W_Q T_i = \begin{pmatrix} 0_d \\ 0_{d'} \\ 0_{d'} \\ f_{i,k+1} \end{pmatrix} \qquad (43)$$

Then softmax attention with this setup yields the output for this softmax attention yields

$$\Omega = \begin{pmatrix} 0_d & \ldots & 0_d & 0_d \\ 0_{d'} & \ldots & 0_{d'} & 0_{d'} \\ \mathbb{E}(w_e)_{|f_{1,k+1}} & \ldots & \mathbb{E}(w_e)_{|f_{N,k+1}} & \mathbb{E}(w_e)_{|f_{N+1,k+1}} \\ 0_{d'} & \ldots & 0_{d'} & 0_{d'} \end{pmatrix} \qquad (44)$$

Via the skip connection for this attention layer, we have

$$T + \Omega = \begin{pmatrix} x_1 & \ldots & x_N & x_{N+1} \\ w_{e,y_1} & \ldots & w_{e,y_N} & 0_{d'} \\ \mathbb{E}(w_e)_{|f_{1,k+1}} & \ldots & \mathbb{E}(w_e)_{|f_{N,k+1}} & \mathbb{E}(w_e)_{|f_{N+1,k+1}} \\ f_{1,k+1} & \ldots & f_{N,k+1} & f_{N+1,k+1} \end{pmatrix} \qquad (45)$$

## E. Details on Generation of the Synthetic Data in the Main Paper

We consider synthetic data generated by the model $p(Y = c|f(x)) = \exp[w_c^T f(x)] / \sum_{c'=1}^C \exp[w_{c'}^T f(x)]$, for $C = 25$ and $w_c \in \mathbb{R}^5$. For data synthesis, the elements of the embedding matrix $W_e \in \mathbb{R}^{5 \times 25}$ are generated (once) randomly, with each matrix component drawn i.i.d. from $\mathcal{N}(0,1)$. After $W_e$ is so drawn, different contextual datasets consider a distinct function $f^{(l)}(x)$, where $l$ represents the context index. To constitute $f^{(l)}(x)$, 5 categories are selected uniformly at random from the dictionary of $C = 25$ categories. Let $c^{(l)}(1), \ldots, c^{(l)}(5)$ denote these categories for context $l$. We further randomly generate 5 respective "anchor positions," $\tilde{x}(1), \ldots, \tilde{x}(5)$, each drawn i.i.d. from $\mathcal{N}(0_d, I_d)$, where $d = 10$ (for covariates $x \in \mathbb{R}^1 0$). The function for context $l$ is represented as

$$f^{(l)}(x) = \lambda \sum_{m=1}^5 w_{c(m)} \kappa_{RBF}[x - \tilde{x}(m); \sigma_m], \qquad (46)$$

where the RBF kernel parameter $\sigma_m$ for component $m$ is selected such that $\kappa_{RBF}[x - \tilde{x}(m); \sigma_m] = \exp(-\sigma_m^2 \|x - \tilde{x}_m\|_2)$ equals 0.1 at the center of the other kernel to which it is closest (in a Euclidean distance sense). We set $\lambda = 10$ (selected so as to have category $c(m)$ be clearly most probable in the region of $\tilde{x}(m)$). Note that the attention-based inference assumes constant attention-kernel parameters for all keys/queries, and therefore these is a model mismatch, in that $f^{(l)}(x)$ responsible for data generation employs a different RBF kernel for the five components in.

## F. Lower-Dimensional Dataset for Further Understanding of Trained TF and Connection to GD

The following additional experiments consider data for which the covariates are two-dimensional. These experiments are considered so as to provide enhanced visualization of the results. Further, because they are of small scale, they allow explicit examination of the degree of match between the Trained TF learned parameters and the associated GD theory.

Specifically, we consider data for which $d = 2$, $C = 20$, and $d' = 5$. Two-dimensional covariates ($d = 2$) allow visualization of predictions across the entire covariate space, and relatively small $C$ and $d'$ are also leveraged to aid interpretation.

The data are generated using (1) from the main paper, with $W_e$ shared across all contexts, and with $f(x)$ that is context-dependent. Matrix $W_e$ is constituted by drawing each of its elements iid from $\mathcal{N}(0,1)$. For $W_e$ so generated, a full set of contextual data $\{\mathcal{C}^{(l)}\}_{l=1,L}$ are generated. When training the models (Trained TF or GD), five different random initializations are used for the model parameters (using initialization methods as described in Glorot & Bengio (2010)). While results are shown here for one $W_e$, performance was found to be consistent across many different draws of $W_e$ (shared across contexts).

The 2D space of covariates $x$ is $[-1,1] \times [-1,1]$, with each of the two components of $x_i$ drawn uniformly over [-1,1]; covariates were drawn similarly in (von Oswald et al., 2023). To define $f(x)$ associated with a particular set of contextual data, the space of covariates is divided into four quadrants, and for a given context $\mathcal{C}^{(l)}$ one of the $C = 20$ classes is associated with each quadrant (selected uniformly at random, without replacement). Let $c_q^{(l)} \in \{0, \ldots, C\}$ be the category associated with quadrant $q = 1, \ldots, 4$, for context $l$. If $x_i$ is in quadrant $q$, we set $f(x_i) = w_{e,c_q^{(l)}}$, where $w_{e,c_q^{(l)}}$ is column $c_q^{(l)}$ of $W_e$. Using $W_e$ and $f(x_i)$ so

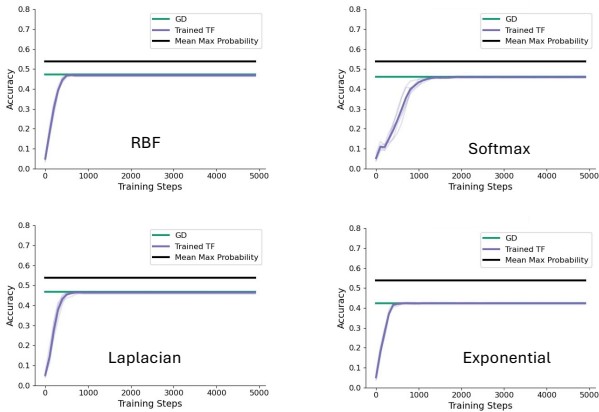

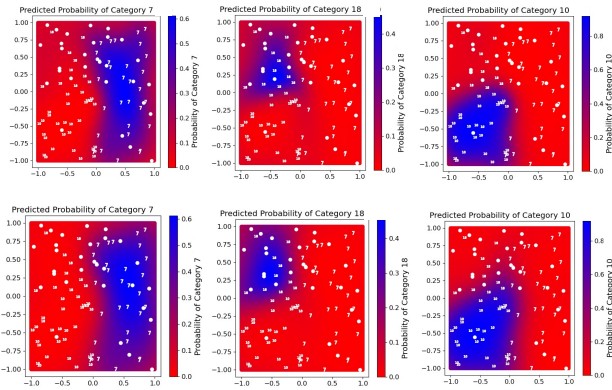

*Figure 8.* For one attention layer, comparison of accuracy of GD and Trained TF, for attention based on RBF, softmax, Laplacian and exponential attention. The horizontal axes reflect Trained TF learning steps.

*Figure 9.* Depiction of $p(Y = c|X = x)$ for queries $x$ considered across the entire 2D space of covariates $x$. Results are for one attention layer and RBF attention, with the top row GD and the bottom Trained TF. The left, middle and right columns consider probabilities $p(Y = c|X = x)$ for $c = 7$, $c = 18$ and $c = 10$, respectively. Contextual samples are depicted in white, where samples of categories 7, 10 and 18 are shown with respective numbers, and all other categories are depicted as circles.

constituted, category $y_i^{(l)}$ for $x_i^{(l)}$ is drawn as in (1) from the main paper.

For each contextual data $\mathcal{C}^{(l)}$, we expect the associated $f(x)$ to have a nonlinear, step-like pattern linked to the quadrants. Note that within the data-generation process we have not explicitly defined $f(x)$ in terms of a function tied to any of the kernels (which is why, in the above, it was not expressed $f_{\psi^{(l)}}(x)$). However, for a particular type of kernel attention, in the forward pass of the Transformer the model effectively infers a context-dependent $f_{\psi^{(l)}}(x)$ approximation to the function, tied to the particular contextual data $\mathcal{C}_l = \{(x_i^{(l)}, y_i^{(l)})\}_{i=1,N}$ observed and to the kernel used.

We consider linear, radial basis function (RBF), exponential, Laplacian and softmax attention; each of these is a kernel (with the reproducing property), except for softmax. Like (Cheng et al., 2024), we show results for softmax attention, for comparison; we show below that softmax attention performs well, consistent with the other nonlinear attention kernels considered. With the exception of the linear kernel, each kernel has a single parameter, that is learned based on $\{\mathcal{C}^{(l)}\}_{l=1,L}$. We consider $N = 100$ samples in each $\mathcal{C}^{(l)}$, and $L = 2048$ contextual sets are used for training. We present performance averaged over post-training contexts $\{\mathcal{C}^{(l)}\}_{l=L+1,L+K}$, for $K = 2048$ and performance evaluated for the prediction connected to the query, given the context.

In Figure 8 we show a comparison of GD and Trained TF for one attention layer, for RBF, softmax, Laplacian and exponential attention. Five curves are presented for GD and Trained TF, corresponding to 5 different parameter initializations, and in most cases they overlap, although multiple Trained TF curves are evident for softmax attention (the bold curve represents the average across the 5 training

runs, and the lighter curves represent individual runs). These curves represent the frequency with which the true $y_{N+1}^{(l)}$ was the most-probable prediction for query $x_{N+1}^{(l)}$.

In Figure 8 the "mean max probability" is the average of the probability of the most-probable category from (1) in the main paper, averaged over all queries, and using the *true* underlying $W_e$ and the true stepped form of $f(x)$ employed when generating the data. This may be viewed as an upper bound on performance, and the capacity to achieve this bound is influenced by the context size (here $N = 100$). The relatively low performance bound (less than 0.6) is due to setting $d' = 5$, which leads any column of $W_e$ to have a large inner product with a subset of other columns, yielding high correlation between associated categories.

In Figure 8 we observe close agreement between the performance of the Trained TF and GD for all kernels. Similar agreement between Trained TF and GD was observed for linear attention, but the predictive accuracy was much worse, likely because of the highly nonlinear nature of $f(x)$ (note as well the relative inferior performance of the exponential kernel).

To further compare the predictions of GD and Trained TF, we plot the inferred probability for select categories, for queries covering the full range of the covariate space (here for one example set of contextual data $\mathcal{C}^{(l)}$). For visualization, we consider a large number of queries for the same context, finely sampled across the 2D covariate space. Importantly, in the Transformer implementation, all such queries can be considered *in a single forward pass* of the Transformer (as discussed above). Results are shown in Figure

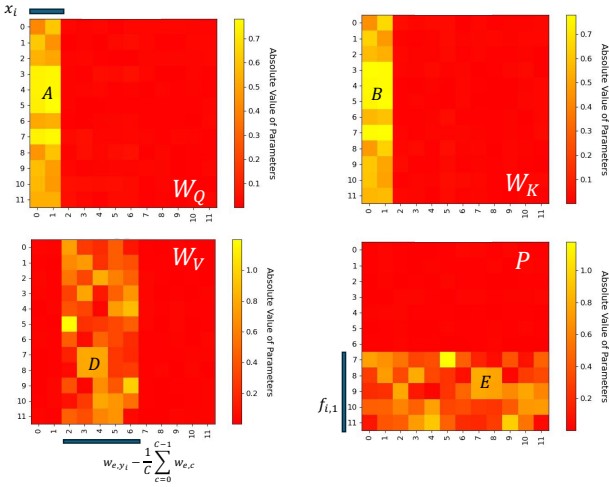

*Figure 10.* Parameters learned for one layer of the Trained TF with softmax attention. Along each axis of the matrices, when applied to the $i$th data sample, elements $(0,1)$ correspond to $x_i \in \mathbb{R}^2$, elements $(2,\ldots,6)$ correspond to $w_{e,y_i} - \frac{1}{C}\sum_{c=1}^{C} w_{e,c} \in \mathbb{R}^5$, and elements $(7,\ldots,11)$ correspond to $f_{i,k} \in \mathbb{R}^5$.

9 for the RBF attention kernel. These results are for one attention layer, considering the probability of 3 (of the 20) categories.

The three categories considered in Figure 9 were selected as follows. Using the true underlying data-generation process, we determine which category is most probable in each quadrant of covariate space. Labeling the quadrants as 1, 2, 3 and 4, starting at the top-right and moving counter clockwise to the bottom right, the respective most-probable categories are $c = 7$, $c = 18$, $c = 10$ and $c = 7$ ($c = 7$ is most probable in two quadrants). In Figure 9, note the close agreement in predictions between GD and Trained TF, and that the high probabilities match the quadrant to which they are linked.

While Figure 8 indicates close agreement in the predictions of GD and Trained TF, further insight is gained by examining the underling parameters learned by Trained TF. Consistency between the learned parameters of the Trained TF and the properties of the GD-imposed parameters would indicate that the Trained TF is learning to perform GD in its forward pass. We focus here on results for one attention layer, as in that case GD-imposed parameters implement the exact update in (4) from the main paper. In Figure 10 we present the learned matrices $W_Q$, $W_K$, $W_V$ and $P$ for softmax attention. The learned Transformer parameters were consistent across all synthesized datasets, and we discuss one representative example.

We first consider $W_Q$ and $W_K$. The input at position $i$ is $e_{i,0} = (x_i, w_{w,y_i} - \frac{1}{C}\sum_{c=1}^{C} w_{e,c}, 0_5)^T$, and via $W_Q$ and $W_K$ in Figure 10 attention is directed only at $x_i$. As predicted by GD, the term $w_{w,y_i} - \frac{1}{C}\sum_{c=1}^{C} w_{e,c}$ is zeroed

out, noting the large blocks of $W_K$ and $W_Q$ that are near zero.

In Figure 10, the sub-matrices, $A$ and $B$ are identified, each in $\mathbb{R}^{12\times2}$. Within the softmax attention, inner products $(Ax_i)^T(Bx_j) = x_i^T Q x_j$ are performed based on the learned form of $W_Q$ and $W_K$, where $Q = A^T B$. Columns 1 and 2 of $A$ are found to be very similar to the respective columns in $B$ (evident by a careful examination of the pixel colors connected to $A$ and $B$ in Figure 10), and columns 1 and 2 of these matrices are found to be almost orthogonal. Consequently, for $A$ and $B$ learned by the Trained TF, $Q \in \mathbb{R}^{2\times2}$ is found to be nearly diagonal, with equal diagonal elements. The value of the diagonals may be seen as a parameter $\lambda$ within the softmax, *e.g.*, as $\exp(\lambda x_i^T x_j)$, and hence the learned diagonal weight connected to $Q$ may be linked to learning the softmax attention parameter. Therefore, we infer that $W_Q$ and $W_K$ learned by the Trained TF are acting as predicted by the GD analysis.

Now consider the form of $W_V$, that has near-zero values on the elements that interact with $x_i$, and hence $W_V$ interacts almost entirely with $w_{w,y_i} - \frac{1}{C}\sum_{c=1}^{C} w_{e,c}$, as predicted by GD. Further, note that matrix $P$ at the output of attention has only its last 5 rows with substantial values, and the last 5 elements of the output is where we anticipate the prediction of $f_{i,1}$ is placed.

In $W_V$ and $P$ of Figure 10 we identify two sub-matrices, $D \in \mathbb{R}^{12\times5}$ and $E \in \mathbb{R}^{5\times12}$. These two sub-matrices are found to satisfy $E \approx D^T$ (evident in the colors in Figure 10). Further, each of the columns of $D$ (rows of $E$) are nearly orthogonal, with approximately the same $\ell_2$ norm. $Q$ and the matrix product $ED$ are shown to be very close to diagonal, with equal diagonal elements.

The updated $f_{j,1}$ manifested by the Trained TF may be expressed as $f_{j,1} \approx ED \sum_{i=1}^{N}[w_{w,y_i} - \bar{w}_e]\kappa(x_i, x_j)$. Since $ED \approx EE^T \approx \gamma I_5$, where $\gamma$ is a scalar and $I_5$ is a $5 \times 5$ identity matrix, we have that the Trained TF represents $f_{j,1} \approx \gamma \sum_{i=1}^{N}[w_{w,y_i} - \bar{w}_e]\kappa(x_i, x_j)$, and this approximation for $f_{j,1}$ is placed in the last 5 components of the attention output. This is exactly how the GD model implements (4) in the main paper, assuming $\gamma$ plays a role like $\alpha/N$.

In the context of Figure 10, we now examine in detail the matrices $W_Q$, $W_K$, $W_V$ and $P$ learned by the Trained TF. In that figure, submatrices $A$, $B$, $D$ and $E$ were identified. In the left of Figure 11 we present $Q = A^T B \in \mathbb{R}^{2\times2}$, which ideally is a diagonal matrix with constant diagonal elements. Results are shown here for one of the five $W_e$ considered for data generation; results were similar for all data considered.

Similarly connected to those results, in the right side of Figure 11 we show $ED \in \mathbb{R}^{5\times5}$, which also should be a diagonal matrix with constant diagonal elements, if the

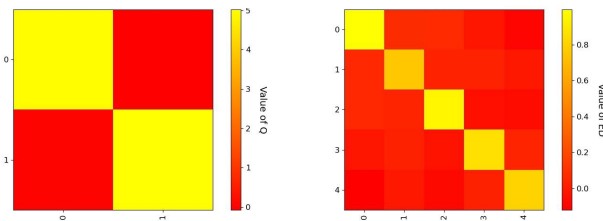

*Figure 11.* Example of learned $Q \in \mathbb{R}^{2 \times 2}$ (left) and learned matrix product $ED \in \mathbb{R}^{5 \times 5}$ (right). These are based on the softmax attention, but all kernels provided similar results.

learned parameters are to be consistent with Trained TF performing a GD step in its forward pass.

The results in Figure 11 provide support to the perspective that the Trained TF, based on a single layer of attention, learns to perform one step of GD in its forward pass.

## G. Details on the gradient updates for real $Y$ and Gaussian $p(Y|X = x)$

We use the notation $f_{j,k} = f_k(x_j)$ to represent the function of interest after the $k$th step of gradient descent, evaluated at covariates $x_j$. For real-valued $Y$ and Gaussian $p(Y|X = x)$ (Ahn et al., 2024; von Oswald et al., 2023; Cheng et al., 2024; Ahn et al., 2023), we have

$$\log p(Y_i = y_i | f(x_i)) = -\frac{1}{2\sigma^2} \|y_i - f(x_i)\|^2 \quad (47)$$

Assuming $\sigma$ is a constant, gradient descent for the parameters $A$ associated with the function $f(x) = A\psi(x)$ yields

$$f_{j,k+1} = f_{j,k} + \underbrace{\frac{\alpha}{N} \sum_{i=1}^{N} (y_i - f_{i,k})\kappa(x_i, x_j)}_{\Delta f_{j,k}} \quad (48)$$

$$= f_{j,k} + \frac{\alpha}{N} \sum_{i=1}^{N} [y_i - \sum_{k'=0}^{k} \Delta f_{i,k'}]\kappa(x_i, x_j) \quad (49)$$

where we typically initialize as $f_{j,0} = 0_{d'}$, where $0_{d'}$ is a $d'$-dimensional vector of all zeros. Consequently, we also have $\Delta f_{i,0} = 0_{d'}$. The variance $\sigma^2$ is treated as a constant, and absorbed into the learning rate (effectively assuming that the variance $\sigma^2$ may be approximated as a constant for all contextual data of interest (von Oswald et al., 2023; Cheng et al., 2024)).

To make the connection to more-general $Y$ and $p(Y|f(x))$, we may re-express (49) as

$$f_{j,k+1} = f_{j,k} + \frac{\alpha}{N} \sum_{i=1}^{N} (y_i - \mathbb{E}(Y)_{|_{f_{i,k}}})\kappa(x_i, x_j)$$

$$= f_{j,k} + \frac{\alpha}{N} \sum_{i=1}^{N} [y_i - \sum_{k'=0}^{k} \Delta_{\mathbb{E}_{i,k'}}]\kappa(x_i, x_j)$$

where $\Delta_{\mathbb{E}_{i,k'}} = \Delta f_{i,k'}$. As indicated in (49), $\Delta f_{i,k'}$ is the direct result of the attention mechanism.

Real $Y$ and Gaussian $p(Y|f(x))$ is a special case, for which $\mathbb{E}(Y)_{|_{f_{i,k}}}$ is simply the latent function $f_{i,k}$. More generally, and of relevance for the categorical data considered here, the needed expectation iss a *nonlinear* function of $f_{i,k}$.

## H. Multiple Kernels for Function Modeling

Assume that we have a latent function

$$f(x) = \sum_{m=1}^{M} A_m^T \psi_m(x) \quad (50)$$

where each feature transformation $\psi_m(x)$ has an associated kernel $\psi_m(x_i)^T \psi_m(x_j) = \kappa(\Lambda_m x_i, \Lambda_m x_j)$, where $\Lambda_m$ is a diagonal matrix that places emphasis on different components of the covariates $x$. The set of feature maps or (equivalently) kernels are fixed, and the set $\{A_m\}_{m=1,M}$ are context-dependent.

Extending the discussion from the paper, employing functional GD, the update equation connected to the $m$th function, $m \in \{1, \ldots, M\}$, is $f_{m,k+1}(x) = A_{m,k+1}\psi_m(x)$, we have

$$f_{m,k+1}(x)$$
$$= f_{m,k}(x) + \frac{\alpha_m}{N} \sum_{i=1}^{N} [w_{e,y_i} - \mathbb{E}(w_e)_{|_{f_k(x_i)}}]\kappa(\Lambda_m x_i, \Lambda_m x_j)$$

The total function is expressed as $f_{k+1}(x) = \sum_{m=1}^{M} P_m f_{m,k+1}(x)$, where each $P_m \in \mathbb{R}^{d' \times d'}$. The learned matrices $\{P_m\}_{m=1,M}$ are consistent with the standard way of merging multiple attention heads in the Transformer (Vaswani et al., 2017).

In practice, the $M$ kernels are connected to $M$ self-attention heads. The use of the inferred function within (1) from the paper remains unchanged, and therefore the expectation $\mathbb{E}(w_e)_{|_{f_{i,k}}}$ is computed unchanged from the paper, using the cross-attention layer.

# I. Theorem 2 and Proof

In this section, we present the formal version of the informal Theorem 1 in Section 5 of the main paper. In Section I.1, we present the key setup and notation for Theorem 2, which is similar to that of Section D. In Section I.2, we present a number of key assumptions.

## I.1. Setup

For $\ell \in 0...L$, we let $Z_\ell \in \mathbb{R}^{(d+d'+d')\times(N+1)}$ denote the input to the $\ell^{th}$ layer of the Transformer. For convenience, let us define $X_\ell$ as the first $d$ rows of $Z_\ell$, $\Gamma_\ell$ as the next $d'$ rows of $Z_\ell$, and $F_\ell$ as last $d'$ rows of $Z_\ell$, i.e. $Z_\ell = \begin{bmatrix} X_\ell \\ \Gamma_\ell \\ F_\ell \end{bmatrix}$. The "readout of the Transformer's prediction for $f(x_i)$ at layer $k$", referenced in Theorem 1, is thus given by the $i^{th}$ column of $f_{i,k}$.

By the above definition, $Z_0$ is the input to the $L$-layer Transformer:

$$Z_0 = \begin{bmatrix} x_1 & x_2 & \cdots & x_N & x_{N+1} \\ w_{e,y_1} & w_{e,y_2} & \cdots & w_{e,y_N} & 0_{d'} \\ 0_{d'} & 0_{d'} & \cdots & 0_{d'} & 0_{d'} \end{bmatrix}.$$

Let $\xi : \mathbb{R}^d \to \mathbb{R}^{d'} \times \mathbb{R}^{d'} \to \mathbb{R}^{d'}$ be defined as follows: for any $f \in \mathbb{R}^{d'}$,

$$\xi(f) = \frac{1}{C} \sum_{c=1}^{C} \frac{\exp(w_{e,c}^\top f)}{1/C * \sum_{c=1}^{C} \exp(w_{e,c'}^\top f)} = \mathbb{E}_{c|f}[w_{e,c}].$$

Let $\tilde{h} : \mathbb{R}^{(N+1)\times(N+1)}$ denote the column-wise activation (with 0 on the last column, to mask out the query). To be specific, each column of $\tilde{h}$ corresponds to one of the $N + 1$ queries, and each row of $\tilde{h}$ corresponds to one of the $N$ keys. In this analysis, it is assumed that the attention, represented as $\tilde{h}(\cdot)$, is a function of $X_0^\top B X_0$. The softmax attention is consistent with that, and hence this analysis extends beyond RKHS attention to softmax attention. In the context of RKHS kernels, linear attention may also be expressed in the assumed form, as can RBF-based attention, assuming that the covariates are normalized for constant $\ell_2$ norm:. Concretely:

$$\tilde{h}_{softmax}(K) := \mathtt{norm}(M \exp(K)), \qquad \tilde{h}_{RBF}(K) := M \exp(K), \qquad \tilde{h}_{linear}(K) := MK \tag{51}$$

where $\exp(K)$ denotes element-wise normalization, $M = \begin{bmatrix} I_{N\times N} & 0_{N\times 1} \\ 0_{1\times N} & 0_{1\times 1} \end{bmatrix}$, and $\mathtt{norm}(\cdot)$ normalizes each column of the input matrix to sum to 1. Note that $\tilde{h}$ plays a role like $\kappa$ in the main paper. In the case of kernels, we consider the special case of kernels for which the resulting function is dependent on the set of inner products $\{x_i^T x\}_{i=1,N}$, and it generalizes to other functions that are so dependent, such as softmax.

Let $\Xi(X, \Gamma, F) : \mathbb{R}^{(N+1)\times d} \to \mathbb{R}^{(N+1)\times d'} \times \mathbb{R}^{(N+1)\times d'} \to \mathbb{R}^{(N+1)\times d'}$ denote the **column-wise** application of $\xi$, i.e.

$$[\Xi(F)]_i = \xi([F]_i) \qquad i = 1...N$$
$$[\Xi(F)]_i = 0 \qquad i = N+1,$$

where in the above, $[\cdot]_i$ denotes the $i^{th}$ column. The function $\Xi(F_\ell)$ computes the expectation $\mathbb{E}(w_e)_{|f_\ell(x)}$ for covariates $x$, for all covariates connected to the context, as in Eq (6) of the main paper. It is assumed that this expectation can be calculated exactly; we show in Section 3.2.2 how this is implemented with cross-attention. This theory evaluates the optimality of the multi-layered self-attention mechanism from the GD perspective, given the assumed exact computation of $\Xi(F_\ell)$ (which we *can* implement exactly within the attention model).

Having defined all the above we define the forward pass of the Transformer as

$$Z_{\ell+1} = Z_\ell + W_{V,\ell} \begin{bmatrix} X_\ell \\ \Gamma_\ell - \Xi(F_\ell) \\ F_\ell \end{bmatrix} \tilde{h}(Z_\ell^\top W_{K,\ell}^\top W_{Q,\ell} Z_\ell). \tag{52}$$

We highlight that (52) differs from the construction in Section D. This is once again because, in 52, we assume access to $\Xi$ which computes $\mathbb{E}(w_e)_{|_{f_\ell(x)}}$, whereas the construction in Section D details how $\Xi$ may be implemented exactly via a cross-attention layer. Our proof generalizes easily to the setting of Section D, but writing things in terms of $\Xi$ significantly simplifies presentation.

Another minor difference is that $W_{V,\ell}$ in 52 is equivalent to $PW_{V,\ell}$ in (22) and (23). This is once again to reduce notation and does not affect the generality of the proof.

Finally, under the above notation, the cross-entropy loss as

$$\bar{\mathcal{L}}(\theta) = -\mathbb{E}_{X_0,\Gamma_0,y_{N+1}} \left[ \langle w_{e,y_{N+1}}, [F_L]_{N+1} \rangle - \log(\sum_{c=1}^{C} \exp(\langle w_{e,c}, [F_L]_{N+1} \rangle)) \right],$$

where $\theta$ denotes the collection of parameters.

We verify that the cross entropy training loss is equivalent to

$$\bar{\mathcal{L}}(B(t,R)) = -\mathbb{E}_{X_0,\Gamma_0,y_{N+1}}[g(X_0,\Gamma,y_{N+1},B(t,R))].$$

## I.2. Assumptions

**Assumption 1.** First, we assume that $W_{K,\ell}^\top W_{Q,\ell}$ satisfies the following sparsity pattern

$$W_{K,\ell}^\top W_{Q,\ell} = \begin{pmatrix} B_\ell & 0_{d\times d'} & 0_{d\times d'} \\ 0_{d'\times d} & 0_{d'\times d'} & 0_{d'\times d'} \\ 0_{d'\times d} & 0_{d'\times d'} & 0_{d'\times d'} \end{pmatrix},$$

where $B_\ell \in \mathbb{R}^{d\times d}$. We will use $B := (B_0, B_1...B_L)$ to denote the tuple of $B_\ell$'s of all layers.

In words, the effect of Assumption 1 enforces that each self-attention module only computes the self-similarity matrix using the feature vectors (columns of $X_\ell$). The embedding vectors $\Gamma_\ell$ and function estimate $F_\ell$ are not involved in the computation of self similarity.

This assumption is deemed justified because $w_{e,y_{N+1}}$ is unavailable for the query, and hence attention will focus on the covariates, which *are* observed for the query, in the form of $x_{N+1}$. The values only act on the same inputs as connected to the keys ($i = 1,\dots,N$) for which embedding vectors $w_{e,y_i}$ are available, and it is these that are tied to the language; therefore we assume $W_V$ acts on the $\{w_{e,y_i}\}_{i=1,N}$ that encode the tokens connected to the context.

**Assumption 2.** We assume that $W_{V,\ell}$ only updates the function estimates $F_\ell$:

$$W_{V,\ell} = \begin{pmatrix} 0_{d\times d} & 0_{d\times d'} & 0_{d\times d'} \\ 0_{d'\times d} & 0_{d'\times d'} & 0_{d'\times d'} \\ 0_{d'\times d} & T_\ell & 0_{d'\times d'} \end{pmatrix}$$

Furthermore, we assume that $T_\ell = t_\ell I$ **is a scaling of the identity matrix, where** $t_\ell$ **is a scalar.** This enforces that the gradient update to each coordinate of the function estimates (each row of $F_\ell$) have the same learning rate.

**Assumption 3.** We assume that $X_0$ (which is the collection of $N+1$ input covariates) has distribution that is rotationally invariant. In other words, $Pr(X_0) = Pr(UX_0)$, where $Pr$ denotes the density or probability of $X_0$.

**Assumption 4.** We make a *distributional* assumption that all there exists a true latent function $f_*(\cdot; X_0)$ from which all the class labels $y_1...y_{N+1}$ are drawn (recall that $X_0$ denotes the collection of input feature vectors). In other words, we assume that for $c \in \{1...C\}$,

$$Pr(y_i = c) \propto \exp(w_{e,c}^\top f_*(x_i; X_0)).$$

Furthermore, We will assume that

$$[f_*(x; X_0)]_j = [\tilde{g}(X_0^\top x)]_j, \tag{53}$$

where $\tilde{g} : \mathbb{R}^{N+1} \to \mathbb{R}^{N+1}$ is any vector-valued function.

One motivating example for assuming the form of $f_*$ is the following:

$$\tilde{g}_j(X_0^\top x) = \sum_{i=1}^{N} \alpha_{ij} \gamma_{ij}(x_i^\top x),$$

where $\gamma_{ij} : \mathbb{R} \to \mathbb{R}$ are arbitrary functions. This definition of $\tilde{g}$ is motivated by the representation theorem (Schölkopf et al., 2001), when $\gamma_{ij}(x_i^\top x_j) = \kappa(x_i, x_j)$ for some kernel $\kappa$. In this case the function is in the RKHS family, as in most of the paper. However, (53) is a more general representation, and allows expansion functions like softmax.

### I.3. Theorem Statement

**Theorem 2 (Formal Version of Theorem 1)**
*Consider the setup in I.1 and assumptions in I.2. There exists a stationary point of the cross-entropy loss $\bar{\mathcal{L}}(\theta)$, where $[F_\ell]_i$ (the $i^{th}$ column of $F_\ell$) evolves (as $\ell$ increases) according to the gradient descent sequence in (2)-(3).*

### I.4. Proof of Theorem 2

Under the sparse structure assumed on $W_V$ and $W_K^\top W_Q$ in Assumptions 1 and 2 above, we verify that for all $\ell$,

$$X_\ell = X_0 := \begin{bmatrix} x_1 & x_2 & \cdots & x_N & x_{N+1} \end{bmatrix}$$
$$\Gamma_\ell = \Gamma_0 := \begin{bmatrix} w_{e,y_1} & w_{e,y_2} & \cdots & w_{e,y_N} & 0_{d'} \end{bmatrix}.$$

and

$$F_{\ell+1} = (I + T_\ell)(\Gamma_0 - \Xi(F_\ell))\tilde{h}(X_0^\top B_\ell X_0) \tag{54}$$

Next, notice that when $B_\ell = b_\ell I$ (where $b_\ell$ are scalars) for all $\ell$, the Transformer exactly implements gradient descent. This is because (54) then simplifies to

$$F_{\ell+1} = (1 + t_\ell)(\Gamma_0 - \Xi(F_\ell))\tilde{h}(b_\ell X_0^\top X_0)$$
$$\Leftrightarrow \quad [F_{\ell+1}]_i = (1 + t_\ell)(w_{e,y_i} - \mathbb{E}(w_e)_{|[F_{\ell+1}]_i})\tilde{h}(b_\ell X_0^\top X_0) \tag{55}$$

Notice that (55) above matches (2)-(3) up to naming of constants. $\tilde{h}$ in (55) generalizes the role of $\kappa$ in (3) (see discussion following (51)).

Therefore, it remains to verify that **there exists a stationary point of $\bar{\mathcal{L}}$ under which $B_\ell = b_\ell I$ for all layers $\ell$.**.

For the rest of this poof, it will be useful to explicitly write down the dependency of $F_\ell$ as a function of $X_0, \Gamma_0, B$:

$$F_{\ell+1}(X_0, \Gamma_0, B) = (I + T_\ell)(\Gamma_0 - \Xi(F_\ell(X_0, \Gamma_0, B)))\tilde{h}(X_0^\top B_\ell X_0) \tag{56}$$

We will also treat the parameters $B$ as variables, and treat all other parameters as constants, i.e. $\bar{\mathcal{L}}(\theta) =: \bar{\mathcal{L}}(B)$.

Our proof will proceed as follows:

1. Let $S_\ell(0) = s_\ell I_{d \times d}$ for all $\ell$ be some arbitrary initialization of $B$. Let $S(0) = (S_1(0)...S_L(0))$

2. Let $S(t) =$ be defined by the dynamics

$$\frac{d}{dt} S_\ell(t) = -\frac{1}{d} \texttt{Tr}(\nabla_{B_\ell} \bar{\mathcal{L}}(S(t))). \tag{57}$$

We claim that for $S(t)$ as defined above,

$$\frac{d}{dt} \bar{\mathcal{L}}(S(t)) \leq -\sum_{\ell=0}^{L} \left\| \nabla_{B_\ell} \bar{\mathcal{L}}(S(t)) \right\|_F^2, \tag{58}$$

where $\| \cdot \|_F$ denotes the Frobenius norm.

3. Notice that under (57), and under the assumed initialization, $S_\ell(t)$ are scalings of $I_{d \times d}$ for all $t$. Since $\bar{\mathcal{L}}$ is lower bounded, (58) must then imply that $\sum_{\ell=0}^{L} \left\| \nabla_{B_\ell} \bar{\mathcal{L}}(S(t)) \right\|_F^2 \to 0$ as $t \to 0$. Furthermore, the limit of $S(t)$ consists of scaled-identity matrices, and thus implements gradient descent as explained above.

It only remains to verify (58), which we do in the next subsection.

### I.5. Proof of (58)

Let $k$ be some fixed but arbitrary layer. Let $R \in \mathbb{R}^{d \times d}$ denote a fixed but arbitrary matrix. Let $B_\ell := b_\ell I_{d \times d}$ for some arbitrary scalars $b_\ell$.

$$B(t, R) := \{B_0, B_1, B_2 ..., B_{k-1}, B_k + tR, B_{k+1} ... B_L\}, \tag{59}$$

i.e. in $(B(t, R))_\ell$ is equal to $(B(0, R))_\ell = B_\ell$ at every layer $\ell$, except for layer $k$, where $(B(t, R))_k = B_k + tR$.

We first verify the following three key facts:

1. For any orthogonal $U \in \mathbb{R}^{d \times d}$ and $h : \mathbb{R}^{d' \times (N+1)} \to \mathbb{R}$,

$$\mathbb{E}_{(\Gamma_0, w_{e}, y_{n+1}) | X_0}[h(\Gamma_0, w_{e, y_{n+1}})] = \mathbb{E}_{(\Gamma_0, w_{e}, y_{n+1}) | U X_0}[h(\Gamma_0, w_{e, y_{n+1}})].$$

   For any $i = 1 ... N + 1$, let $[X_0]_i = x_i$ be the $i^{th}$ column of $X_0$, then

$$Pr(y_i = c) \propto \exp(w_{e,c}^\top f_*([X_0]_i; X_0)).$$

   By definition of $f_*$ in (53), $f_*([UX_0]_i; UX_0) = \tilde{g}(X_0^\top U^\top U [X_0]_i) = \tilde{g}(X_0^\top [X_0]_i) = f_*([X_0]_i; X_0)$

   The conclusion then follows by noticing that for each $i = 1 ... N + 1$, $y_i$'s are independent, conditioned on $x_i$'s.

2. For any $\ell$, and for any $X_0, \Gamma_0$,
$$F_\ell(X_0, \Gamma_0) = F_\ell(U X_0, \Gamma_0, B)$$

   We will prove this by induction. The base case is $F_0 = 0$, which holds trivially. Suppose the above holds for some $\ell$. By (56),

$$\begin{aligned} F_{\ell+1}(U X_0, \Gamma_0, B) &= (I + T_\ell)(\Gamma_0 - \Xi(F_\ell(U X_0, \Gamma_0, B))) \tilde{h}(X_0^\top U^\top B_\ell U X_0) \\ &= (I + T_\ell)(\Gamma_0 - \Xi(F_\ell(X_0, \Gamma_0, B))) \tilde{h}(X_0^\top B_\ell X_0) \\ &= F_{\ell+1}(X_0, \Gamma_0, B). \end{aligned}$$

   In the above, the second equality is by the inductive hypothesis.

3. Let $J_{\tilde{h}}(A, C) : \mathbb{R}^{(N+1) \times (N+1)} \times \mathbb{R}^{(N+1) \times (N+1)} \to \mathbb{R}^{(N+1) \times (N+1)}$ denote the Jacobian of $\tilde{h}$, i.e. $\frac{d}{dt} \tilde{h}(A + tC) \big|_{t=0} = J_{\tilde{h}}(A, C)$.

   We will prove by induction the following: for all $\ell$,

$$\frac{d}{dt} F_\ell(U X_0, \Gamma_0, B(t, R)) \Big|_{t=0} = \frac{d}{dt} F_\ell(X_0, \Gamma_0, B(t, U^\top R U)) \Big|_{t=0}. \tag{60}$$

   For all $\ell \le k$, we verify by definition that

$$\frac{d}{dt} F_\ell(U X_0, \Gamma_0, B(t, R)) \Big|_{t=0} = 0 = \frac{d}{dt} F_\ell(X_0, \Gamma_0, B(t, U^\top R U)) \Big|_{t=0}. \tag{61}$$

This is because $F_\ell$ depends only on $B_i$ in layers $0...\ell$, which do not change with $t$. Next, for layer $k+1$,

$$
\begin{aligned}
&\frac{d}{dt}F_{k+1}(UX_0, \Gamma_0, B(t, R))\Big|_{t=0} \\
=&(I+T_k)(\Gamma_0 - \Xi(F_k(UX_0, \Gamma_0, B(0, R))))\frac{d}{dt}\tilde{h}(X_0^\top U^\top(B_\ell + tR)UX_0)\Big|_{t=0} \\
=&(I+T_k)(\Gamma_0 - \Xi(F_k(UX_0, \Gamma_0, B(0, R))))J_{\tilde{h}}(X_0^\top U^\top B_\ell UX_0, X_0^\top U^\top RUX_0) \\
=&(I+T_k)(\Gamma_0 - \Xi(F_k(X_0, \Gamma_0, B(0, R))))J_{\tilde{h}}(X_0^\top B_\ell X_0, X_0^\top U^\top RUX_0) \\
=&(I+T_k)(\Gamma_0 - \Xi(F_k(X_0, \Gamma_0, B(0, R))))\frac{d}{dt}\tilde{h}(X_0^\top(B_\ell + tU^\top RU)X_0)\Big|_{t=0} \\
=&\frac{d}{dt}F_{k+1}(X_0, \Gamma_0, B(t, U^\top RU))\Big|_{t=0}
\end{aligned}
$$

where the first equality is by chain rule and (61).

Finally, consider $\ell \geq k+1$. Let $J_\Xi(A, C) : \mathbb{R}^{(N+1)\times d} \times \mathbb{R}^{(N+1)\times d} \to \mathbb{R}^{(N+1)\times(d)}$ denote the Jacobian of $\tilde{h}$, i.e. $\frac{d}{dt}\Xi(A + tC)\Big|_{t=0} = J_\Xi(A, C)$.

$$
\begin{aligned}
&\frac{d}{dt}F_{\ell+1}(UX_0, \Gamma_0, B(t, R))\Big|_{t=0} \\
=&(I+T_k)(-\frac{d}{dt}\Xi(F_\ell(UX_0, \Gamma_0, B(t, R)))\Big|_{t=0})\tilde{h}(X_0^\top U^\top B_\ell UX_0) \\
=&(I+T_k)(-J_\Xi(F_\ell(UX_0, \Gamma_0, B(0, R)), \frac{d}{dt}F_\ell(UX_0, \Gamma_0, B(t, R))\Big|_{t=0})))\tilde{h}(X_0^\top B_\ell X_0) \\
=&(I+T_k)(-J_\Xi(F_\ell(X_0, \Gamma_0, B(0, R)), \frac{d}{dt}F_\ell(X_0, \Gamma_0, B(t, U^\top RU))\Big|_{t=0})))\tilde{h}(X_0^\top B_\ell X_0) \\
=&(I+T_k)(-\frac{d}{dt}\Xi(F_\ell(X_0, \Gamma_0, B(t, U^\top RU)))\Big|_{t=0})\tilde{h}(X_0^\top B_\ell X_0) \\
=&\frac{d}{dt}F_{\ell+1}(X_0, \Gamma_0, B(t, U^\top RU))\Big|_{t=0}
\end{aligned}
$$

The first equality is by chain rule, the second equality is by definition of $J_\Xi$. The third equality is by inductive hypothesis, and by item 2 above. This concludes the proof.

Let us define $g$ as follows:

$$
g(X_0, \Gamma_0, y_{N+1}, B(t, R)) = \langle w_{e, y_{N+1}}, [F_L(X_0, \Gamma_0, B(t, R))]_{N+1}\rangle - \log(\sum_{c=1}^{C}\exp(\langle w_{e, c}, [F_L(X_0, \Gamma_0, B(t, R))]_{N+1}\rangle)),
$$

where $B(t, R)$ is as defined in (59) above for some arbitrary matrix $R$. As a consequence of item 3 above, we verify that

$$
\frac{d}{dt}g(UX_0, \Gamma_0, y_{N+1}, B(t, R))\Big|_{t=0} = \frac{d}{dt}g(X_0, \Gamma_0, y_{N+1}, B(t, U^\top RU))\Big|_{t=0} \tag{62}
$$

Then

$$\frac{d}{dt}\bar{\mathcal{L}}(B(t,R))\Big|_{t=0}$$

$$= -\mathbb{E}_{X_0,\Gamma_0,y_{N+1}}\left[\frac{d}{dt}g(X_0,\Gamma_0,y_{N+1},B(t,R))\Big|_{t=0}\right]$$

$$= -\mathbb{E}_{X_0}\left[\mathbb{E}_{\Gamma_0,y_{N+1}|X_0}\left[\frac{d}{dt}g(X_0,\Gamma_0,y_{N+1},B(t,R))\Big|_{t=0}\right]\right]$$

$$= -\mathbb{E}_{X_0,U}\left[\mathbb{E}_{\Gamma_0,y_{N+1}|UX_0}\left[\frac{d}{dt}g(UX_0,\Gamma_0,y_{N+1},B(t,R))\Big|_{t=0}\right]\right]$$

$$= -\mathbb{E}_{X_0,U}\left[\mathbb{E}_{\Gamma_0,y_{N+1}|UX_0}\left[\frac{d}{dt}g(X_0,\Gamma_0,y_{N+1},B(t,U^\top RU))\Big|_{t=0}\right]\right]$$

$$= -\mathbb{E}_{X_0,U}\left[\mathbb{E}_{\Gamma_0,y_{N+1}|X_0}\left[\frac{d}{dt}g(X_0,\Gamma_0,y_{N+1},B(t,U^\top RU))\Big|_{t=0}\right]\right]$$

$$= -\mathbb{E}_{X_0}\left[\mathbb{E}_{\Gamma_0,y_{N+1}|X_0}\left[\frac{d}{dt}g(X_0,\Gamma_0,y_{N+1},B(t,\mathbb{E}_U[U^\top RU]))\Big|_{t=0}\right]\right]$$

$$= -\mathbb{E}_{X_0,\Gamma_0,y_{N+1}}\left[\frac{d}{dt}g(X_0,\Gamma_0,y_{N+1},B(t,\mathbb{E}_U[U^\top RU]))\Big|_{t=0}\right]$$

$$= \frac{d}{dt}\bar{\mathcal{L}}(B(t,\mathbb{E}_U[U^\top RU]))\Big|_{t=0}$$

The second equality is by iterated expectation.

In the third equality, $U$ is uniformly sampled random orthogonal matrix. This equality holds because we assume that $P(X) = P(UX)$.

In the fourth equality, we use (62).

In the fifth equality, we use item 1 above, with $h(\Gamma_0, w_{e,y_{n+1}}) := \frac{d}{dt}g(X_0,\Gamma_0,y_{N+1},B(t,U^\top RU))\Big|_{t=0}$.

In the sixth equality, we use the fact that $\mathbb{E}_{\Gamma_0,y_{N+1}|X_0}\left[\frac{d}{dt}g(X_0,\Gamma_0,y_{N+1},B(t,\mathbb{E}_U[U^\top RU]))\Big|_{t=0}\right]$ depends on $U$ only via $U^\top RU$, and is linear in $U^\top RU$.

The seventh equality is again by iterated expectation.

Now let $B_\ell := S_\ell(t)$, where $S_\ell(t)$ is as defined in (57), and let $R := -\nabla_{S_\ell(t)}\bar{\mathcal{L}}(S(t))$. Then

$$-\left\|\nabla_{B_k}\bar{\mathcal{L}}(S(t))\right\|_F^2$$

$$= -\left\langle\nabla_{B_k}\bar{\mathcal{L}}(S(t)),\nabla_{B_k}\bar{\mathcal{L}}(S(t))\right\rangle$$

$$= \left\langle\nabla_{B_k}\bar{\mathcal{L}}(S(t)),R\right\rangle$$

$$= \frac{d}{dt}\bar{\mathcal{L}}(B(t,R))\Big|_{t=0}$$

$$= \frac{d}{dt}\bar{\mathcal{L}}(B(t,\mathbb{E}_U[U^\top RU]))\Big|_{t=0}$$

$$= \left\langle\nabla_{B_k}\bar{\mathcal{L}}(S(t)),\mathbb{E}_U[U^\top RU]\right\rangle$$

$$= \left\langle\nabla_{B_k}\bar{\mathcal{L}}(S(t)),\frac{d}{dt}S_k(t)\right\rangle,$$

where the last equality uses (57) together with the fact that $\mathbb{E}_U[U^\top RU] = \frac{1}{d}\text{Tr}(R)$.

Since $k$ is arbitrary, we conclude that

$$\frac{d}{dt}\bar{\mathcal{L}}(S(t)) = \sum_{k=0}^{L}\left\langle\nabla_{B_k}\bar{\mathcal{L}}(S(t)),\frac{d}{dt}S_k(t)\right\rangle = -\sum_{k=0}^{L}\left\|\nabla_{B_k}\bar{\mathcal{L}}(S(t))\right\|_F^2,$$

which proves (58).

```
In the following exercise, the student is given a pre-written beginning of a story.
The student needs to complete this story.
The exercise tests the student´s language abilities and creativity.

Here is the pre-written beginning:

<PROVIDED BEGINNING>
**STORY_BEGIN**
</PROVIDED BEGINNING>

And here is the students response:

<STUDENT RESPONSE>
**STORY_END**
</STUDENT RESPONSE>

First, provide a concise qualitative assessment about the student's writing.
Then, give the writing a grade out of 10.
These assessments should be done for each of the following rubric items:

1. Grammar:
* Is the writing grammatically correct?
* Evaluate syntax, punctuation, and sentence structure.
2. Consistency:
* Is the student's writing consistent with the provided beginning of the story?
* How well does the student complete the final sentence of the prescribed beginning?
3. Plot:
* Does the plot of the student's writing make sense (regardless of the provided beginning)?
4. Creativity:
* How creative is the student's writing?

Format your response as follows:

<GRAMMAR>
[Qualitative assessment of grammar]
</GRAMMAR>
<GRAMMAR_GRADE>
[Grade out of 10]
</GRAMMAR_GRADE>

<CONSISTENCY>
[Qualitative assessment of consistency]
</CONSISTENCY>
<CONSISTENCY_GRADE>
[Grade out of 10]
</CONSISTENCY_GRADE>

<PLOT>
[Qualitative assessment of plot]
</PLOT>
<PLOT_GRADE>
[Grade out of 10]
</PLOT_GRADE>

<CREATIVITY>
[Qualitative assessment of creativity]
</CREATIVITY>
<CREATIVITY_GRADE>
[Grade out of 10]
</CREATIVITY_GRADE>

Provide your assessment below:
```

*Figure 12.* The user prompt given to GPT-4o to generate the model scores.

```
You are a writing evaluator designed to assess student story completions.
You will be provided children's stories written for a 3-4 year old audience.
Your role is to provide constructive, fair, and detailed evaluations based on
specific rubric criteria.
```

*Figure 13.* The System prompt given to GPT-4o.

## J. Language Model Evaluation Details

Our quantitative evaluations used the OpenAI API with the GPT-4o model. For each story, GPT-4o was given the user prompt in Figure 5, with the `**STORY_BEGIN**` keyword replaced with the true beginning of the story (used as the evaluated model's input) and the `**STORY_END**` keyword replaced with the generated ending to the story. The numeric scores were then parsed from the GPT-4o model responses. Additionally, the system prompt in Figure 6 was provided for all evaluations.

## K. Additional Language-Model Experiments

In the main paper, we considered only softmax-based attention to allow for a more direct comparison to the Transformer model. Here we include results for the linear and RBF kernels. In addition, we provide results for the linear,

| Model | Grammar | Consistency | Plot | Creativity | Overall |
|---|---|---|---|---|---|
| True Ending | 9.21 | 8.77 | 7.77 | 6.68 | 8.11 |
| Linear GD | 4.77 | 3.64 | 3.30 | 3.81 | 3.88 |
| RBF GD | 3.83 | 2.99 | 2.72 | 3.32 | 3.21 |
| Softmax GD | 4.25 | 3.40 | 3.08 | 3.76 | 3.62 |
| Linear GD (FF) | 5.94 | 4.57 | 4.20 | 4.44 | 4.79 |
| RBF GD (FF) | 5.11 | 4.09 | 3.72 | 4.07 | 4.24 |
| Softmax GD (FF) | 5.90 | 4.55 | 4.22 | 4.70 | 4.84 |
| Transformer | 5.97 | 4.98 | 4.43 | 4.75 | 5.04 |

*Figure 14.* GPT-4o scoring of generated story endings for all models. Each item is graded out of a maximum score of 10.

---

### Linear GD Model (Good Example)

"One day, a bossy cat and a nice dog were playing a sport. They were kicking a ball. The cat was very bossy and wanted to win. The dog said, "Let's play and have fun. We can both be good!" The cat did not like this idea. The cat wanted to be the best . The cat said, "Okay, let's play together." They played and had fun. But then, something unexpected happened. A big wind came and blew the ball far away. The cat and the dog tried to catch the ball, but they could not catch it. Then, something unexpected happened. A big wind came and took the ball away. The cat and the dog were very surprised. They did not know what to do. They tried to catch it, but they could not "

### Linear GD Model (Bad Example)

"Once upon a time, there was a little girl called Lucy. She really liked playing with the board. She had lots of fun with it every day. One day, Lucy was playing with her board in her backyard. Suddenly, she heard a loud noise. It was the delivery truck. Lucy yelled as she spat, "Oh no, the delivery is here!" Lucy frantically ran inside and quickly grabbed her board. She wanted to keep it away from the delivery driver. But, it was too late. The delivery driver saw her and yelled, "Hey, give me!" Lucy was so happy to have a new friend. The next day. She said, "Let's go!" Lucy was so excited to see the package. She said, "It's okay!" Lucy was so happy. She thanked her mom and said, "It's okay." Lucy smiled. She said, "It's okay." "

---

*Table 4.* Generations from the GD model with linear attention

RBF, and softmax GD models with the addition of the feed-forward element, as found in the transformer model. While this deviates from gradient descent, it allows for an even closer comparison with the transformer.

### RBF GD Model (Good Example)

"Once upon a time, in the city of Columbus, there was a professional soccer team called the Columbus Crew. This team had lots of dedicated fans who loved watching them play every week. One day, two best friends named Alex and Jamie deci ded to go to a Columbus Crew game together. They were so excited because they heard that the team had something extra special planned for their fans! As soon as they arrived at the stadium, they received free t-shirts and hats with the Columbus Crew logo on them. "Wow!" said Alex, putting on his new hat. "This is so cool! I love getting free stuff." Alex and Jamie decided to visit the local library. As they walked, they noticed something strange happening around them. There were many people wearing masks, wearing masks and gloves, hats, gloves and masks. Mr. Johnson explained, "These masks can help protect our bodies need to protect us from harmful germs. We need to protect our masks and protect ourselves against harmful germs." Feeling inspired, Alex and Jamie decided to spread awareness throughout the neighborhood. Soon enough, "

### RBF GD Model (Bad Example)

"Once upon a time, there was a small bird named Tim. Tim was very proud of his language. He could talk and sing better than any other bird in the forest. One day, Tim met a big cat named Max. Max was not nice. He liked to demand things f rom others. Max said to Tim, "Sing for me, little bird." Tim was scared, but he sang his best song for Max. Max was happy with Tim's song, but he wanted more. "Sing again," Max demanded. Tim sang again, but he was tired and sad. He wished he could find a way to stop Max from the tree. The moral of the story is to be kind to help others. The moral of the story is to be kind to help others. The moral of the story is to be kind to help others. The moral of the story is to be kind to help others. The moral of the story is to be kind to help others. The moral of the story is to be kind to help others. The moral of the story is to be kind to help others "

*Table 5.* Generations from the GD model with RBF attention

### Linear GD Model with Feed-Forward (Good Example)

"One day, a little girl named Lily wanted to play dress-up. She went to her room and looked at all her costumes. She saw a big box with a new costume inside. Lily was very happy and opened the box fast. Lily took out the costume and examined it. It was a superhero costume with a red cape. She put it on and said, "Now I can be a superhero!" Lily ran around the house, laughing and having fun. But then, something unexpected happened. The costume started to talk! It said, "Thank you, Lily! I am a magic superhero! I can grant you one wish." Lily was so surprised and happy. She wished for a new friend to play with. From that day on, Lily and the magic fairy became best friends. They had lots of fun together. And they lived happily ever after. The moral of the story is to always "

### Linear GD Model with Feed-Forward (Bad Example)

"Once upon a time, there was a lonely tree. The tree had one olive. The olive was green and small. A little girl named Lily lived near the tree. Every day, Lily would go to the tree and gaze at the olive. One day, Lily saw a bird on the tree. The bird looked at the olive too. Lily said, "Hi bird, do you like the olive?" The bird said , "Yes, I can help you." Lily and the bird became good friends. They played together every day. They had lots of fun. And they lived happily ever after. The moral of the story is to be kind and help others when they need it. And they lived happily ever after. The moral of the story is to always be kind and help others when they need it. The moral of the story is to always be kind and help others when they need it "

*Table 6.* Generations from the GD model with linear attention (with the feed-forward)

### Softmax GD Model with Feed-Forward (Good Example)

"Once upon a time, in a town called Melodyville, there lived two best friends named Rhythm and Harmony. They loved exploring different types of music from all around the world! One day they heard that the annual Musicland Awards were ha ppening soon, and they wanted to learn more about what made these musicians so special. Rhythm asked their teacher, Mr. Tempo, "How does the Musicland Academy choose the winners?" Mr. Tempo explained, "Well, just like in real life, many things play a part - including cultural context and political climate."Harmony was curious, so she asked, " How do we know if we can find out more?" Mr. Rhythm explained, "Well my dear friends, it depends on various factors such as sound waves, frequency, amplitude, and pitch. When we listen to each other, it creates sound waves that travel through the air and reach our ears." Melody added, "That sounds amazing! But how does it work?" Mr. Melody smiled and replied, "That's right! Sound waves are made up of tiny "

### Softmax GD Model with Feed-Forward (Bad Example)

"Once there was a generous little girl named Amy. She wanted to go to the fountain so she could play. She zipped in her red shoes and ran towards the playground. When she got to the fountain, she saw two boys splashing around. Amy asked if she could play too, but the boys said no. They were not being very generous. Instead , Amy decided to share her toys with her friends. They all played together and had lots of fun. The moral of the story is that it's important to be kind and share with each other. The moral of the story is to always be kind and share with each other. The moral of the story is to always be kind and share with your friends. The moral of the story is to always be kind and share with others. The moral of the story "

*Table 7.* Generations from the GD model with softmax attention (with the feed-forward)

