# OpenReview forum: "On Understanding Attention-Based In-Context Learning for Categorical Data"
_ICML.cc/2025/Conference — ICML 2025 poster_

### Official Review · Reviewer_Hsdu · 2025-03-12

**Overall Recommendation:** 3

**Summary:**

This paper investigates how transformers perform in-context learning on categorical data, extending prior work that has largely focused on in-context regression tasks.
The authors provide a kernel-based functional gradient descent perspective, wherein each layer of the transformer can be interpreted as performing one step of gradient-based inference using in-context examples.
They first demonstrate an expressivity result: that attention-based architectures can implement kernel-based gradient descent for categorical inference.
They further prove that their idealized construction corresponds to a stationary point in the loss landscape.
Moreover, they empirically validate their framework, showing that trained models converge to solutions that closely align with their theoretical predictions.
To support their findings, they conduct experiments on datasets such as ImageNet, Tiny Stories, and Children Stories to demonstrate the predictive power of their approach across both language and vision tasks.

**Claims And Evidence:**

Yes, it seems so.

**Essential References Not Discussed:**

Not that I'm aware of.

**Experimental Designs Or Analyses:**

The experiments (Section 6) were not clear to me, and I would appreciate clarification on the following points:
* I understand that you are comparing two models: "Trained TF" and "GD" (it would be good to re-introduce them here, despite the mentions in Section 4). Could you please clarify how "GD" is constructed --- you would need access to some $\psi$ to make $f_\phi$, and how is this $\psi$ derived/learned?
* I'd appreciate more detail + exposition on why you chose to generate synthetic data in this way (Section 6.1)
* I don't understand the problem setup for in-context image classification (Section 6.2). Could you please clarify what an example task might look like?
* I also don't understand the language modeling experiment (Section 6.3). Similarly, could you please provide more clarification on example tasks --- in addition to those in Table 2? Could you please make the connection between in-context classification and text generation more explicit?

**Methods And Evaluation Criteria:**

Yes, I think so.

**Other Comments Or Suggestions:**

N/A

**Other Strengths And Weaknesses:**

Strengths:
* The theoretical contributions look good, and I appreciate the mathematical rigor!


Weaknesses:
* I had trouble getting through the notation, particularly in Section 2. However, because these notations are so important to the paper, I implore the authors to generously use diagrams and examples. Some discussions and experiments may be added to the appendix if space is lacking.
* As mentioned above, some experiment descriptions were not clear to me. For example, in-context image classification is not a "natural" task for me, so I would appreciate some examples --- even if they may be toy.


Overall, I found this to be an interesting read and a solid paper, but I believe that its clarity could be refined.

**Questions For Authors:**

See above.

**Relation To Broader Scientific Literature:**

This expands on our understanding of in-context learning, particularly for categorical data. Much of the literature focuses on regression, so looking at in-context classification is quite important.

**Theoretical Claims:**

Yes, they make sense to me. I checked some proofs in Appendix B and C.

---

> ### Author Rebuttal · Authors · 2025-03-30
>
> Thank you for your careful review and for your valuable feedback. On your questions:
>
> 1. "Trained TF" and "GD" (we agree that it would be good to re-introduce them here, despite the mentions in Section 4; we will do that).
> The detailed derivation of the GD updated equation is in Appendix B and summarized in Eq (13). This is functional GD in an RKHS space for the latent function, based on cross-entropy loss for that function (with the loss minimized via functional GD, effected in the Transformer inference stage, based on the observed contextual data). In (20) we show the form of the $W_Q$, $W_K$ matrices, and for these there are no learnable parameters -- within a permutation, these matrices come directly from GD theory. In (21) we show $W_V$, and for that there is one learnable parameter, learning rate $\alpha$. There is in general a different learning rate at each layer. In addition, for GD we learn the category-dependent embedding vectors $\\{w_{e,c}\\}\_{c=1,C}$. By contrast, in Trained TF all parameters are learned, without restrictions (including the category-dependent embedding vectors). In Figs 8 and 9 of the Appendix we show close agreement (on a low-D example, for interpretation) between Trained TF and GD parameters, indicating (as supported by our theory) that the Trained TF does indeed learn to perform inference like functional GD. However, Fig 4 shows that Trained TF needs a lot of data to achieve such learning, while GD-based composition of the model parameters trains well with far less data (as it has far fewer parameters).
>
> 2. Synthetic data generation: In Appendix E we provided more details. In short, there is a latent, context-dependent function $f(x)$, which the Transformer infers (at the $N$ sample points), in context. To generate synthetic data, we first synthesize the embedding vector $\\{w_{e,c}\\}\_{c=1,N}$, with each component of each vector drawn from $N(0,1)$. These vectors drive the softmax on categories, given $f(x_i): SM(f(x))=\exp[f(x)^Tw_{e,c}]/\sum_{c^\prime} \exp[f(x)^Tw_{e,c^\prime}]$. With {w_{e,c}} set, we generate $f(x)$ as a summation of kernels, to yield a highly nonlinear function (kernel centers positioned at random). Then, for $N$ samples, we draw $N$ covariate vectors $\\{x_i\\}\_{i=1,N}$, from N(0,I). We stick each $x_i$ into $f(x)$ to yield $f(x_i)$, and this goes into the $SM(f(x))$ over categories. We draw a category $y_i\sim SM(f(x_i))$. Now we have $\\{(x_i,y_i)\\}\_{i=1,N}$ which is the context sent into the Transformer, along with query $x_{N+1}\sim N(0,I)$. Each contextual set has a uniquely (randomly) generated underlying $f(x)$.
>
> 3. In-context image classification. Consider context that is characterized by 5 image types (classes) that have never been seen by the ICL model. In the contextual data, we are given 10 examples images of each of the 5 image types (e.g, type of dog, type of cat, etc). So we have 50 contextual samples, each one of 5 classes. The image is represented in feature space by an arbitrary (but good) feature representation. Here we have used VGG features from a CNN, but they could also be from a ViT. For image $i$ let the features be represented as $x_i\in R^d$. For the contextual samples, we also have the label $y_i\in\\{1,\dots,5\\}$. We have five classes, but the details of those labels and image types is arbitrary. Key is that the ICL model has never seen these image classes before. The ICL is given context=$\\{(x_i,y_i)\\}\_{i=1,50}$, manifested in terms of 10 (random) examples for each of the 5 classes. It is also given a query $x\_{N+1}$ from one of the 5 image classes, and is asked to label it, or more specifically $p(Y=c|X=x_{N+1},\text{context})$, for $c=1,\dots,5$.
>
> 4. Language modeling. In-context classification has $C$ classification labels. For language, the $C$ labels correspond to all the (discrete) tokens ($C$ is the number of unique token types). For ICL, we have context $\\{(x_i,y_i)\\}\_{i=1,N}$, x_i covariates and y_i category (token) label. In ICL, the category/token is encoded by a learned embedding vector $w_{e,y_i}$ for token $y_i$, just like in NLP. To model language, we use the positional embedding vector to represent the covariates $\\{x_i\\}\_{i=1,N}$, and also $x_{N+1}$. With this mapping ICL can be used directly for language modeling.
>
> As recommended by you, we will add a figure in the main body of the paper, to help with readability. Please see here [\[LINK\]](https://anonymous.4open.science/r/tmprepo-4447/transformer_rebuttal_additional_plots.pdf) a draft figure (which we will seek to further improve) to aid readability and provide clarity.

---

### Official Review · Reviewer_GFKD · 2025-03-15

**Overall Recommendation:** 1

**Summary:**

The paper seems to be dealing with learning the implicit function embedded in the examples within prompt in in-context learning setting.

The paper is poorly written and difficult to understand. The introduction does not clarify what the proposed method is aiming at, there are disconnected/disparate components stated in methods section that make it very difficult to follow. The experimental setting and description are equally murky.

More details below

**Claims And Evidence:**

First of all, is this a theoretical treatment of function learning during in-context training or a more practical model proposal to explicitly learn the aforementioned (implicit) functions? The title, Lines 041 ~ 014 next column and section 5 give an impression of theoretical analysis whereas contribution 1(b), 2(a,b) and Section 3 alludes to a new model proposal.

The method description and analyses do not qualify  for a theory paper and the experimental results do not seem to pass the bar of a practical model proposal paper.

**Essential References Not Discussed:**

--

**Experimental Designs Or Analyses:**

Although the paper shows some experimental results on public datasets, the problem setting for Imagenet classification Section 6.2, and Language modeling in Section 6.3 qualifies for a demonstration on a toy problem, not a full fledged evaluation needed for a new proposed model architecture.
But, then again, I am not sure if this is a theoretical analysis paper or not.

**Methods And Evaluation Criteria:**

I am also confused what are the methods section claiming. Is a method to *explicitly* learn the function represented implicitly in the prompt during in context training? Section 2.2, and subsection in Section 3 appear to be designed for explicit learning of this function. Is my understanding correct?

If so, why would we need to learn the function explicitly when multiple studies (Ahn 2023, Zhang 2023) already demonstrates that these implicit functions are learned organically during in context training?

**Other Comments Or Suggestions:**

---

**Other Strengths And Weaknesses:**

--

**Questions For Authors:**

---

**Relation To Broader Scientific Literature:**

--

**Theoretical Claims:**

---

---

> ### Author Rebuttal · Authors · 2025-03-30
>
> Thank you for reviewing our paper. Concerning your comments:
>
> > The introduction ....
>
> - lines 20-21 state that the goal of this paper is "extending the functional GD framework [for ICL], to handle categorical observations".
> - Reviewers 4VEb and Hsdu both provided accurate summaries of the goal of the paper in the "Summary" portion of their review.
>
> > ... disconnected/disparate components ....
>
> - Sec. 2.1: we describe the problem -- the labels y conditioned on covariates x are drawn proportional to $\exp(w_{e,y}^T f_{\phi}(x))$ for some unknown $f_{\phi}$ that the Transformer learns in-context (implicitly).
>
> - Sec. 2.2: we describe the equations for updating $f_{j,k}$ to $f_{j,k+1}$ in a single functional GD step (lines 110-113). The key formula is Eq (3), which consists of two parts: (A) computing $w_{e,y}-\mathbb{E}[w_{e}|f_i]$, and (B) computing a weighted RKHS average $\sum_{i=1}^N (\cdots) κ(x_i,x_j)$.
> - Sec 3.2: we show that a self-attention layer can exactly compute (B): the weighted average $κ(x_i,x_j)$. (line 184), and in Sec 3.3 we show the cross attention can exactly compute the expectation in (A).
>
> >  .... designed for explicit learning of this function. Is my understanding correct?  ....
>
> - This is incorrect. Our goal is also to learn $f_{\phi}$ implicitly.
> - Ahn and Zhang considered **linear** latent functions with **real** observations. This led to a gap between theory and practice, and prior theory has **only been validated on synthetic data**.
> - We address a gap in the literature: Our central contribution is extending existing analysis to categorical data. There are significant complexities, such as the fact that the GD update is no longer linear, and contextual demonstrations are no longer $y_i=f(x_i)$ but rather a random draw $y_i\sim \exp(w_{e,y}^\top f(x_i))$.
>
> > The experimental setting  ....
>
> - Our goal is **not to propose a new model**. We have not alluded to or suggested that the proposed model is superior to existing Transformers.
>     - The model in Sec 3 consists of only 3 components: self attention, cross attention, and skip connections. All of these modules are **standard components of a Transformer**. We noted on line 119 that (Vaswani et al 2017)'s original Transformer architecture also contains interleaved self-attention and cross-attention layers.
>     - In 1(b), when we say "introduce a new attention-based architecture", we mean with respect to the existing linear-Transformer design of previous theoretical papers such as (von Oswald, Ahn, Zhang, Cheng). **While it is true that an important innovation of our paper is the analysis of how cross-attention perfectly facilitates GD with respect to categorical data**, we again emphasize that **cross-attention is a very standard component of modern Transformers**.
>     - 2(a,b) are simply experiments to validate our theory -- that the Transformer construction in Sec. 3 is capable of matching in-context GD for tasks on real-world datasets. We are unsure why this "alludes to a new model proposal".
>
> > ... do not qualify for a theory paper ...
>
> - Can reviewer GFKD please **elaborate on why "the method description and analyses do not qualify for a theory paper"?**
>     - In Secs. 2 and 3, we **show rigorously that our proposed construction indeed implements exactly one step of gradient descent for categorical data**. Proof details are provided in Appendix B,C,D.
>     - In Theorem 1 (also 2), **we prove that our proposed construction is a stationary point of the training loss**.
>     - The strength of our results are no weaker than analogous results in earlier works that analyze multi-layer Transformers on real-valued data.
>
> > ... (Ahn, Zhang) already demonstrates ...
>
> - This is incorrect. Our goal is also to learn $f_{\phi}$ implicitly in the Transformer forward pass, like Ahn/Zhang, but for the first time here for categorical observations.
> - We acknowledged the importance of existing work, such as Ahn and Zhang. However, most practical applications of Transformer networks are over **categorical-valued** data. This led to a gap between theory and practice, and prior theory has **only been validated on synthetic data**. To the best of our knowledge, our experiments in Sec 6 are the **first time** that the ICL theory for Transformers has been validated on **any** real-world tasks and data.
> - The central contribution of this work is extending existing analysis to categorical data. There are significant complexities, such as the fact that the GD update is no longer linear, and contextual demonstrations are no longer $y_i=f(x_i)$ but rather a random draw $y_i\sim \exp(w_{e,y}^\top f(x_i))$.
>
> > ... some experimental results  ...
>
> - The goal of our experiments is **not to propose an architecture change for existing LLMs**. Our experiments **empirically validate our theory on real-world data and tasks** -- something which was **not possible in previous papers** as they **do not handle the setting of categorical-valued observations**.

---

### Official Review · Reviewer_4VEb · 2025-03-19

**Overall Recommendation:** 4

**Summary:**

The paper explores theoretical understanding of the In-Context Learning in Transfomer-stack models while dealing with categorical data.
It attempts to design a transformer block that can do gradient-descent in-context.

Authors try to construct a transformer stack that can, in theory perform ICL on categorical data.
They use a softmax transformer for this. The task is to ‘learn” a function “f” from the context, from categorical data using transformer’s forward pass.
They start by expressing the required GD equation for this function, in terms of output embeddings generated by the transformer
They next show in the special case of the real-valued function (regression tasks), this task is easily handled by simply self-attention. This work has already been published.

This paper then engineers input representations and Q, K, V weight matrices to achieve the above derived GD update equation to “f”.
They go on to show the performance of the GD-enabled transformer stack on synthetic, image and language modeling tasks.

## update after rebuttal
Authors have proposed adding another diagrammatic representation of their tecnnique. That might help fix the fact that the paper is hard to understand and follow. No other update from my end

**Claims And Evidence:**

They claim that the particular structure of transformer stacks suggested in this paper is capable of showing GD-learning type behaviour for in-context data and produce proper probability distributions for categorical class prediction

Their claims are well backed by the experimental results they’ve show with varied objective - synthetic class prediction task, image task and next-token prediction too. The additional heat maps for learned-matrices showing confirming the predicted values for weights is a good cross-check on the overall learning happening in the network

**Essential References Not Discussed:**

NA

**Experimental Designs Or Analyses:**

Experimental setup and evaluation looks sound, if the claims are accurate.

No issues found on my end.

**Methods And Evaluation Criteria:**

Their evaluation strategy for the tasks look good.
They also claim to have taken care to use distinct contextual data for testing the models after training. The evaluation setup is really sound in the sense that the test-time classes are not trained on. New set of classes & corresponding embeddings are provided directly in context and the model appears to give good test-time quality after sufficient training.

**Other Comments Or Suggestions:**

TYPOS:

- in Section 3.2; line 2: "k" in "W_k " needs to be capitalized
- Large number of Typos in Appendix "E" (x \in R^{10}, incomplete sentence in the end, ...)



SUGGESTIONS:
- In appendix, around equation 35: just stating P1, P2 = +-1 might make it clearer
- Param setup with Q, K, V matrix sizes involving 0_{3d} etc look sprung-up on the reader at first-pass. Not sure if there's any way to better justify that, but I couldn't get past without Appendix read in detail.

**Other Strengths And Weaknesses:**

Strengths
Paper has a highly mathematical approach to designing a GD-emulating transformer stack. It’s exhaustive, discuss all details required for understanding the hypothesis and the model setup. The extra material in the Appendix is definitely useful and aids understanding of the parameter setups. Lot of work exists around linear transformers emulating real-valued outputs - this work builds on top of that & extends that understanding to categorical outputs with softmax (albeit with a big constraint on the transformer block)
Categorical outputs are especially useful.

They have made good effort to exhaustively test the methodology on a wide array of tasks and are able to show that their approach manages to emulate learning via GD. The technique has been tested on sufficient number of very different scenarios. The evaluation criteria is pretty strong. The details in Appendix are exhaustive and cover theoretical aspects pretty well.


Weaknesses
Presentation:  The paper is extremely dense.  This is likely because of 8-page limit, but lot of details could have been omitted and the paper could have been made more readable/accessible with some helpful diagrams.

Some important explanations didn’t find place in the main paper. As an example in the same domain, [Akyurek et al] https://arxiv.org/pdf/2211.15661 is lot more readable at around the same length

**Questions For Authors:**

1. Do you have a hypothesis on why we need multiple layers of the transformer block, if a single block can emulate GD perfectly well?

**Relation To Broader Scientific Literature:**

Resources
- Transformers Implement Functional Gradient Descent (Cheng et al): https://arxiv.org/pdf/2312.06528
- Transformers Learn in-context by GD (https://arxiv.org/pdf/2212.07677)
- WHAT LEARNING ALGORITHM IS IN-CONTEXT LEARNING? (https://arxiv.org/pdf/2211.15661)


Went through some papers listed above that analyze implicit learning algorithms implemented by trained transformers for tasks like linear regression.


Normal proof methodology involves showing that:
1. Attention mechanism _can_ emulate learning algorithms for task T

2. the data transformations involved in GD/other learning algorithms for task “T”, etc can be performed by transformers stack. OR, designing Q, K, V weight matrices such that the forward-pass can create equivalent learning outcome

3. Trained transformer stack do seem to emulate learning algorithms on task “T”

4. By creating special data for task “T”, comparing performance of trained transformer stack to GD, etc

This paper uses a special transformer block to make the inference pass behave like a GD update.

**Theoretical Claims:**

The paper is extremely dense in theoretical proofs. They claims and proofs mostly looked sound to me. Pointing out some typos etc in "comments" section. Some errors/gaps may have slipped by me though

---

> ### Author Rebuttal · Authors · 2025-03-30
>
> Thank you for your very careful review and helpful feedback.
>
> We will work to make the paper more readable and understandable in the main body of the paper. We constituted Fig 1 with the goal of trying to summarize the setup in a figure, but we can do more. Please see here [\[LINK\]](https://anonymous.4open.science/r/tmprepo-4447/transformer_rebuttal_additional_plots.pdf) a draft figure we propose adding to the body of the paper, to summarize things and hopefully provide more intuition without getting too much into the technical details. We will revise the final version of the paper, with the goal of enhancing readability. Thank you for looking carefully at the Appendix. As you suggested, we will try to move as many insights from there into the main paper.
>
> We will fix the typos and will implement your suggestions.
>
> On your question: Each attention block implements one step of functional GD. An attention block corresponds to a self-attention layer and a subsequent cross-attention layer, as in Fig 1. Multiple attention blocks implement multiple steps of functional GD. In Fig 3, for example, the multiple layers of attention blocks corresponds to multiple steps of GD.

---

### Decision · Program_Chairs · 2025-05-01

**Decision:**

Accept (poster)

**Comment:**

This paper investigates how transformers perform in-context learning on categorical data. The core idea is to introduce a new attention-based architecture, built upon self-attention and cross-attention blocks, that enables exact implementation of multi-step gradient descent with categorical data. Experimental results and analysis look interesting. As pointed by all reviewers, the paper is dense and can be made more readable. As promised in the rebuttal, I recommend authors to incorporate the diagrams and provide more intuitive explanations through out the paper to help readers follow the work more easily. Overall, I am leaning towards accepting this work.